# Local conditions and policy design determine whether ecological compensation can achieve No Net Loss goals

Laura J. Sonter [1,2✉], Jeremy S. Simmonds [1,2], James E. M. Watson [1,2,3], Julia P. G. Jones [4], Joseph M. Kiesecker[5], Hugo M. Costa[3], Leon Bennun [6], Stephen Edwards[7], Hedley S. Grantham[3], Victoria F. Griffiths [8], Kendall Jones[3], Kei Sochi[5], Philippe Puydarrieux [7], Fabien Quétier [9], Helga Rainer [10], Hugo Rainey[3], Dilys Roe[11], Musnanda Satar[12], Britaldo S. Soares-Filho [13], Malcolm Starkey[6], Kerry ten Kate[14], Ray Victurine[3], Amrei von Hase [14], Jessie A. Wells [1] & Martine Maron [1,2✉]

Many nations use ecological compensation policies to address negative impacts of development projects and achieve No Net Loss (NNL) of biodiversity and ecosystem services. Yet, failures are widely reported. We use spatial simulation models to quantify potential net impacts of alternative compensation policies on biodiversity (indicated by native vegetation) and two ecosystem services (carbon storage, sediment retention) across four case studies (in Australia, Brazil, Indonesia, Mozambique). No policy achieves NNL of biodiversity in any case study. Two factors limit their potential success: the land available for compensation (existing vegetation to protect or cleared land to restore), and expected counterfactual biodiversity losses (unregulated vegetation clearing). Compensation also fails to slow regional biodiversity declines because policies regulate only a subset of sectors, and expanding policy scope requires more land than is available for compensation activities. Avoidance of impacts remains essential in achieving NNL goals, particularly once opportunities for compensation are exhausted.

[1] Centre for Biodiversity and Conservation Science, The University of Queensland, St Lucia, QLD 4072, Australia. [2] School of Earth and Environmental Sciences, The University of Queensland, St Lucia, QLD 4072, Australia. [3] Wildlife Conservation Society, Global Conservation Program, New York, NY 10460, USA. [4] School of Natural Sciences, College of Engineering and Environmental Science, Bangor University, Bangor LL57 2UW, UK. [5] Global Lands, The Nature Conservancy, Fort Collins, CO 80524, USA. [6] The Biodiversity Consultancy, 3E King's Parade, Cambridge CB2 1SJ, UK and Conservation Science Group, Department of Zoology, University of Cambridge, Downing St., Cambridge CB2 3EJ, UK. [7] International Union for the Conservation of Nature, 1196 Gland, Switzerland. [8] Department of Zoology, University of Oxford, Zoology Research and Administration Building, 11a Mansfield Road, Oxford OX1 3SZ, UK. [9] Biotope, 22 Boulevard Maréchal Foch, F-34140, BP 58 Mèze, France. [10] Arcus Foundation, CB1 Business Centre, Leda House, Twenty Station Road, Cambridge CB1 2JD, UK. [11] International Institute for Environment and Development (IIED), London WC1X 8NH, UK. [12] Yayasan Konservasi Alam Nusantara, Jakarta, Indonesia. [13] Centro de Sensoriamento Remoto, Universidade Federal de Minas Gerais, Av. Antônio Carlos 6627, Belo Horizonte - MG CEP 31270-900, Brazil. [14] Forest Trends, Washington, DC 20036, USA. ✉email: l.sonter@uq.edu.au; m.maron@uq.edu.au

Halting biodiversity loss and securing ecosystem services are fundamental challenges facing humanity[1]. While industrial development is often important for pursuing economic goals[2], it places immense pressure on ecosystems[3,4]. In response, many nations have adopted ecological compensation policies[5,6] to address the negative impacts of development projects, often in an attempt to achieve "No Net Loss" (NNL) of biodiversity and other related goals, such as securing the provision of ecosystem services valued by local people. These policies invoke the "mitigation hierarchy", where biodiversity losses from development are first avoided wherever possible, then minimised and remediated and, finally, offset to generate commensurate biodiversity gains elsewhere[7]. Hundreds of compensation policies exist worldwide[8,9], including corporate standards and requirements set by financial institutions; yet, their contribution to conservation goals remains uncertain at global, national and even local scales[10]. Indeed, some compensation policies appear to facilitate ongoing biodiversity losses[11,12] and cause further damage to ecosystem services[13,14].

If ecological compensation is to become a cornerstone of attempts to achieve global conservation goals[15], understanding what leads to policy success (or failure) is vital. Several factors complicate efforts to obtain such insight. First, compensation policies vary enormously in their design[16], making it difficult to pinpoint which factors explain differences in project outcomes. For example, some policies focus on generating biodiversity gains using restoration activities ("Improvement" approaches). Others derive gains by protecting existing biodiversity (such as by creating new or strengthening existing protected areas) on the presumption that this prevents its future loss ("Averted Loss" approaches). Second, the biodiversity gains generated by compensation activities can vary substantially from place to place, even for identical policies. For example, Improvement approaches may be more effective when applied to highly degraded areas located nearby well-functioning ecosystems that help promote recovery. Further, developing realistic counterfactual scenarios to evaluate compensation activities against is challenging because biodiversity trajectories differ from place to place– some areas experience significant loss, others natural recovery. While many other factors likely contribute to policy success, including governance and enforcement capacity, understanding the specific influence of these two factors—policy design and local conditions—will help to improve the outcomes of compensation activities worldwide.

Here, we investigate how policy design and local conditions interact to influence potential policy performance—i.e. how close compensation comes to achieving NNL goals. While previous research has explored specific compensation policies and outcomes in certain locations[12,17–19], ours is the first to systematically examine a common set of policy settings across multiple case studies (Fig. 1), which vary in the local conditions that potentially influence policy performance (Table 1). We bring together the policy designs currently promoted globally[8,9] to examine 18 different options, representing combinations of two area-based approaches to generating biodiversity gains, four types of trades in biodiversity features between development and compensation sites, and three methods for prioritising compensation across the landscape (compensation policies often include a combination of these activities; Fig. 2). Our policy design options are hypothetical, yet they represent those used globally and some also resemble those currently used or proposed in our case study regions (see Supplementary Methods 1–4).

We use spatial simulation models to quantify impacts of both future regulated development projects and the compensation activities they require on biodiversity (using the extent of native vegetation types as a proxy; see Methods section) and two ecosystem services (carbon storage and sediment retention). Our definition of what constitutes regulated development differs among case studies and is based on policy trends and the industries these policies will likely regulate in future (see Methods section). We measure impacts of compensation relative to a counterfactual scenario, representing the unregulated biodiversity losses and gains (i.e. changes in vegetation extent simulated by land use change models) likely to occur in absence of regulated development and compensation activities; and define NNL to occur when the impacts of compensation equal or exceed those of development. The Methods detail our conceptual framework (including a description of the variables being manipulated, measured and compared among case studies), our modelling approach (to simulate regulated development and compensation activities, and measure impacts on biodiversity and ecosystem services), and a discussion of assumptions and potential limitations of our study. Supplementary Methods 1–4 contain case study data, model calibration and validation results. We find that two factors limit potential achievement of NNL of biodiversity and ecosystem services, and even the best performing policies fail to slow regional biodiversity declines.

## Results and discussion

**Overview**. Not one of our investigated compensation policy designs achieved NNL of native vegetation extent, our proxy for biodiversity, yet some policies performed better than others did (Fig. 2). Two local conditions explained differences in policy performance among case studies: the extent of land available for compensation limited our ability to protect or restore biodiversity, while counterfactual biodiversity losses and gains within compensation sites limited their potential impact on biodiversity. However, even our best performing policies did little to slow regional biodiversity declines because compensation was only required for a subset of development sectors and expanding policy scope to regulate other types of development would require more land than would be available for compensation under our simulated development scenarios. While expanding policy scope and increasing our set multipliers (to require more compensation per unit impact of development) are both theoretical responses to policy failure, often neither are possible in practice, and so avoidance of impacts remains key to halting biodiversity declines.

**Performance differs among compensation policy design options**. Our results suggest that performance varied substantially between Averted Loss (i.e. protecting existing vegetation and averting counterfactual losses) and Improvement (restoring and protecting land currently void of vegetation) approaches to generating biodiversity gains (Fig. 2). However, our chosen multipliers—i.e. the amount of compensation required per unit impact of regulated development—were key in explaining these differences. Averted Loss multipliers should be set based on counterfactual biodiversity losses (i.e. those which are avertable), whereas Improvement multipliers should capture restoration uncertainties and discount any counterfactual biodiversity gains[20,21]. However, in practice, policies employ relatively arbitrary multipliers, set according to the perceived feasibility of industry delivering on them[22]. We too chose arbitrary (yet plausible) multipliers to enable comparisons among case studies with differing local conditions. We set an Averted Loss multiplier of four, implying a counterfactual biodiversity loss of 20%, and as is required by policy in two of our case studies[12,17]; and an Improvement multiplier of two, assuming 50% success rate of restoration, based on evidence from the restoration ecology literature[23–25]. These decisions directly affected the performance of each approach: Improvement appeared to out-perform Averted Loss consistently

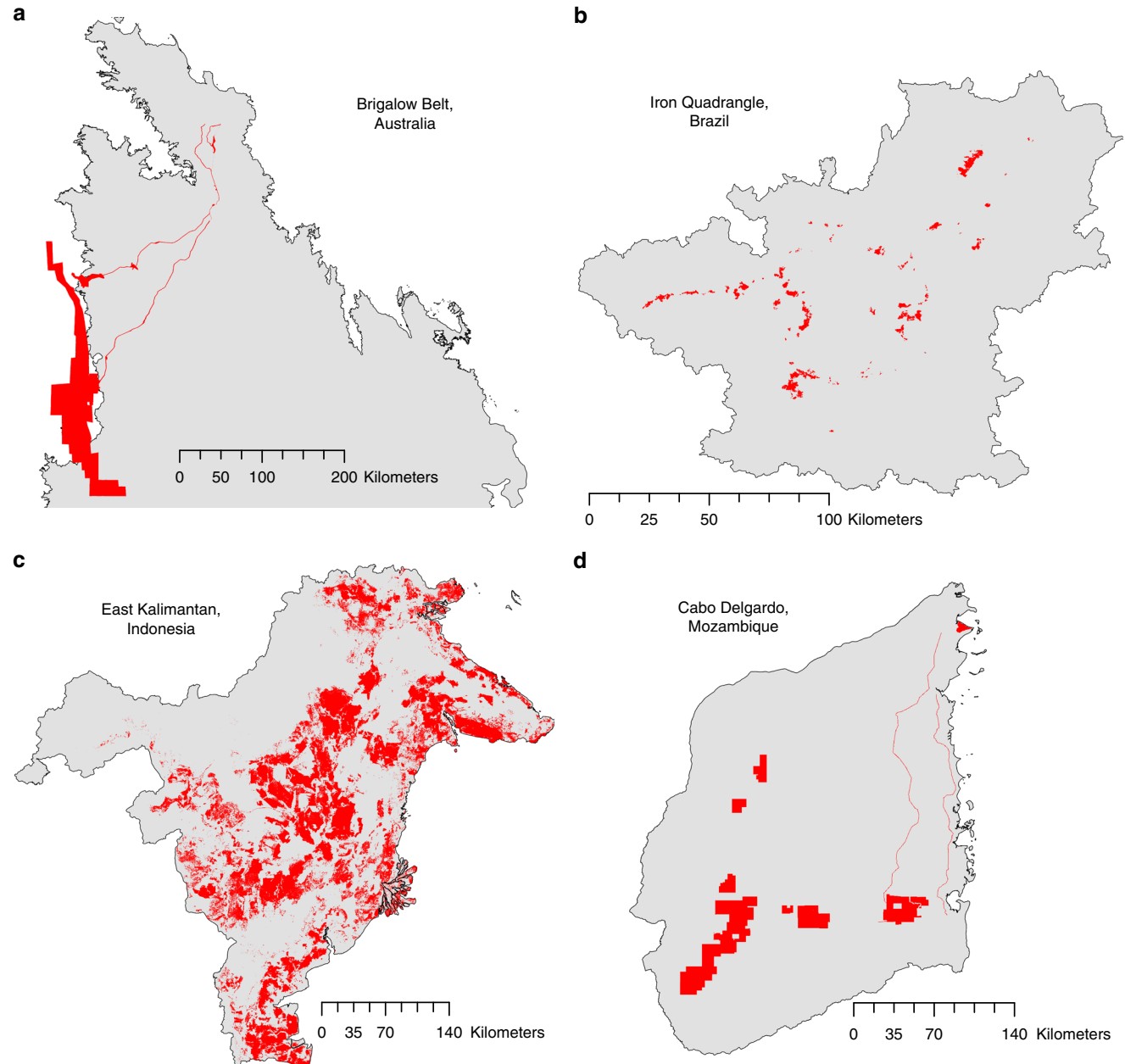

**Fig. 1 Four case studies, which differ markedly in regulated development (shown in red) and local conditions. a** Brigalow Belt, Australia; **b** Iron Quadrangle, Brazil; **c** East Kalimantan, Indonesia; **d** Cabo Delgado, Mozambique. Regulated development represents new mining areas and related infrastructure in the Brigalow Belt and Cabo Delgado; new mining areas in the Iron Quadrangle; and new mining areas and oil palm plantations in East Kalimantan. See Table 1 for summaries of how local conditions differ among cases.

(Fig. 2). Ultimately, our Averted Loss multiplier was too low to achieve NNL because counterfactual biodiversity losses were <20% of the area of compensation and development, while our Improvement multiplier appeared to perform better (because counterfactual biodiversity gains were rare) but largely due to our optimistic (and undifferentiated) assumption of 50% restoration success (see Methods section for further discussion of this assumption). Indeed, Averted Loss approaches may outperform Improvement approaches if rates of restoration success are low and counterfactual biodiversity gains high. Thus, our results for Improvement approaches should not be directly compared with those for Averted Loss approaches and we limit comparisons of performance to within each approach and discuss the multipliers that would be required to achieve NNL in later sections (see "Best performing policies fail to achieve NNL").

Performance varied among the four types of trades in biodiversity features between development and compensation sites (Fig. 2). These trades were done "Out-of-Kind" (where the biodiversity feature gained through compensation does not need to be the same kind as that lost to development), "In-Kind" (where it does), and "Trading-up" (where the biodiversity gained is of higher conservation priority, being either more rare ("Trading-up: Rarity") or at more risk of being lost without intervention ("Trading up: Additional Gains"; note that both forms of "trading-up" can also be considered variations of Out-of-Kind trades). Trading-up: Additional Gains performed best, due to its explicit goal of protecting areas most at risk of counterfactual biodiversity losses (when using Averted Loss) and least likely to undergo counterfactual biodiversity gains (using Improvement), assuming protection comprehensively prevents

**Table 1 Differences in local conditions across case studies.**

| Case study | Region extent (10³ km²) | Biodiversity remaining (%) | Impact of regulated development on biodiversity (km²) | Counterfactual biodiversity loss (km²) | Land available for compensation (km²) | |
| --- | --- | --- | --- | --- | --- | --- |
| | | | | | Improvement | Averted Loss |
| Brigalow Belt, Australia | 26 | 29 | 29 | 225 | 7383 | 7623 |
| Iron Quadrangle, Brazil | 19 | 46 | 56 | 208 | 18,408 | 7276 |
| East Kalimantan, Indonesia | 127 | 80 | 6311 | 5005 | 6408 | 40,514 |
| Cabo Delgado, Mozambique | 77 | 91 | 3051 | 3537 | 551 | 5810 |

Biodiversity remaining indicates the current extent of natural vegetation relative to its pre-clearing extent. Regulated development varies across cases (Fig. 1) and its impact on biodiversity represents the extent of vegetation cleared by these developments during the simulation time period for each case study. Counterfactual biodiversity losses indicate unregulated vegetation clearing during the simulation time period. Land available for each compensation approach is defined as either the extent of clear land available for Improvement, or unprotected vegetation available for Averted Loss. Refer to Supplementary Methods 1–4 for more detail.

any further losses (see Methods section for a discussion of this assumption). In East Kalimantan and Cabo Delgado, Out-of-Kind and Trading-up policies performed equally well when using Improvement because the extent of regulated development and its required compensation resulted in the restoration of all available land. Additionally, when using Averted Loss in East Kalimantan, Trading-up: Additional Gains performed similarly to other trades because most vegetation had a similar chance of being lost. For both approaches, the impacts of In-Kind trades are limited in contexts where insufficient opportunities exist to undertake compensation, or when vegetation types impacted by development are not threatened by other unregulated sectors.

Performance also varied between our three methods for prioritising compensation activities across the landscape: anywhere "Outside PAs", outside but "Near PAs", and strictly "Within PAs" (Fig. 2). Prioritising compensation Outside PAs performed better than Near PAs because areas near protection tended to experience less counterfactual biodiversity losses (thus reducing impacts of Averted Loss) and more gains (reducing impacts of Improvement). The exception was for Cabo Delgado, where prioritising compensation Near PAs performed better, since existing protected areas occur close to development pressures, such as roads and urban centres. Prioritising compensation Within PAs was even less effective given that already protected sites were expected to undergo less counterfactual losses than sites outside them. Only in two case studies—East Kalimantan and Cabo Delgado (when using Improvement)—did protected areas experience some counterfactual losses and contain some cleared land for restoration. However, even then, these impacts were much smaller than those achieved by prioritising compensation elsewhere in the landscape.

Some policies achieved NNL for specific vegetation types (Supplementary Fig. 1), despite failing to achieve NNL overall (Fig. 2). Compensation was most likely to achieve NNL for those vegetation types that would otherwise experience large counterfactual losses (when using Averted Loss) and small counterfactual gains (when using Improvement) relative to impacts of regulated development (Supplementary Methods 1–4). We also found that some policies had large impacts on a few vegetation types, while others had small impacts across many. For example, when using Improvement in the Iron Quadrangle, In-kind trades came within 5% of achieving NNL of all vegetation types, whereas Trading-up (targeting Additional Gains) exceeded NNL by more than 150% for one third of vegetation types (Supplementary Fig. 1).

**Policy performance also depends on local conditions**. We also found that policy performance varied as a result of interactions between policy design and local conditions. In some case studies, policy design options were limited by land availability (Table 1; see Methods section for model assumptions), that is, insufficient unprotected vegetation to protect for Averted Loss, or cleared land to restore for Improvement, to allow compensation for all regulated development simulated over our investigated timeframe. The effect of land availability was most evident in East Kalimantan, where we estimated regulated development (for mining and oil palm expansion) to clear 6311 km² of forest (Table 1) and thus require 12,622 km² of restoration (using Improvement and our multiplier of 2), an area twice that available for restoration (Fig. 3). Land availability was even more influential for less-flexible compensation policies, such as those requiring In-Kind trades. For example, in Cabo Delgado, even though a significant amount of land was available for protection (Table 1), regulated development (for mining and infrastructure expansion) had the largest impact on deciduous miombo savannah woodlands (WSW28; 2008 km²; Supplementary Fig. 1) and this vegetation type had the least available land for restoration, thus compensation (using Improvement) required more than 2.6 times the area available (Fig. 3). In practice, many other factors further limit opportunities to implement compensation policies, such as whether land is actually available at the scale required for NNL. However, our results reveal that to achieve NNL of biodiversity, development impacts on biodiversity must cease once compensation opportunities are exhausted.

Secondly, policy performance reflected the rates of counterfactual biodiversity losses and gains expected within compensation sites. Averted Loss performed well when large counterfactual losses occurred. Such was the case in the Brigalow Belt (Table 1), where extensive vegetation clearing for cattle grazing was expected; however, even then, for every hectare of land cleared by regulated development (i.e. for mining and infrastructure expansion) compensation (using our multiplier of 4) averted less than half a hectare of future loss. Conversely, the performance of policies using Improvement were influenced by counterfactual biodiversity gains rather than losses; restoring land that would have otherwise naturally recovered is not additional and so policy performance was reduced where recovery was more likely. Again, this was most evident in the Brigalow Belt, where natural regrowth and recovery is common and, if left to establish for more than 30 years, can ultimately resemble remnant vegetation structurally and ecologically[17]. In subsequent sections, we discuss

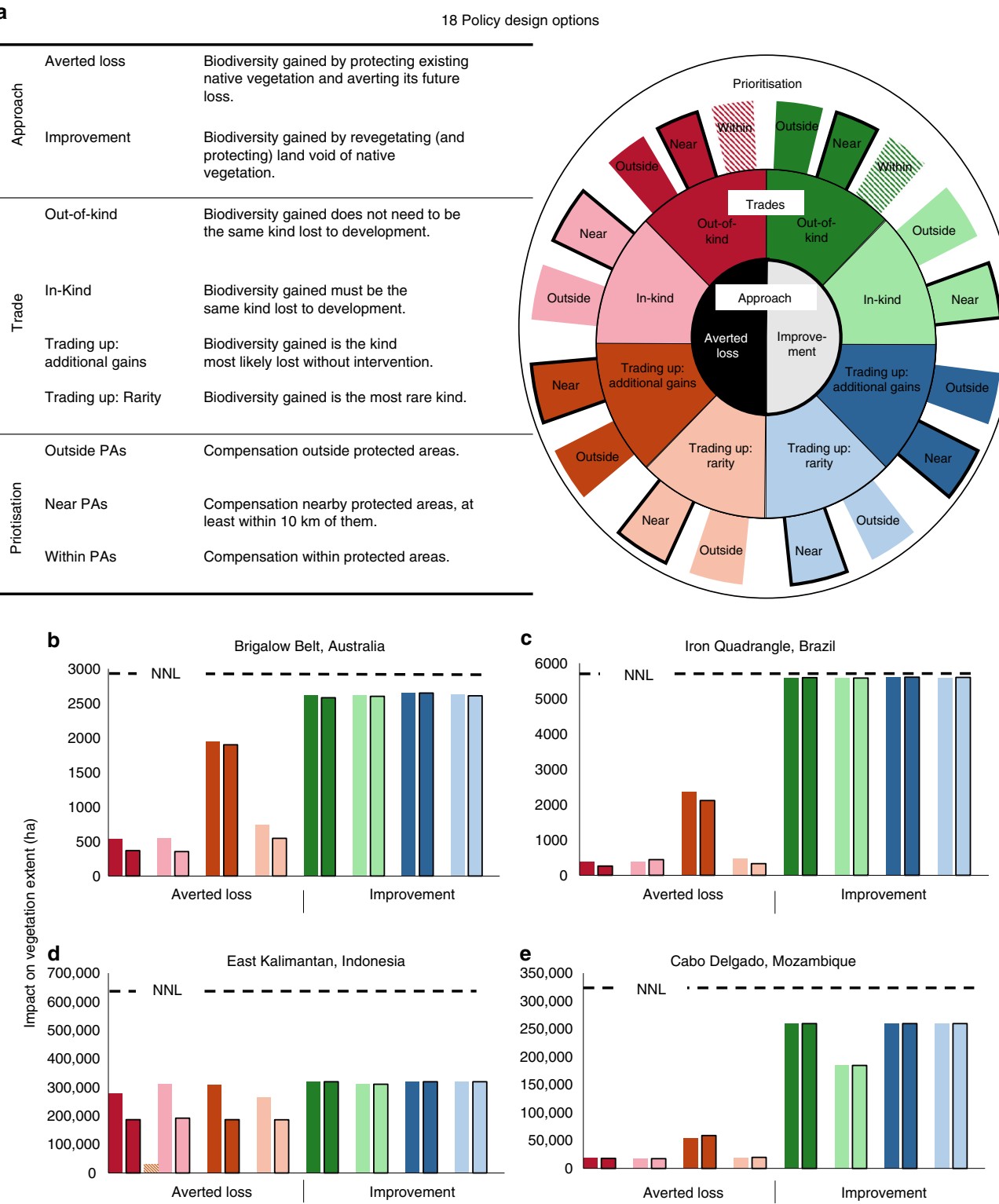

**Fig. 2 Our 18 investgated compensation policy options and their impacts on biodiversity. a** Policy design options representing two area-based approaches to achieving biodiversity gains, four options in trading biodiversity between development and compensation sites, and three methods of prioritising compensation activities to the landscape. Note: we only prioritised compensation Within PAs for Out-of-Kind trades, given that protected areas contained few opportunities for compensation (i.e. they did not contain much cleared land to restore, or experience large counterfactual losses to avert) and even more restrictive 'In-Kind' trades were rarely possible. Graphs **b**–**e** show negative impacts of regulated development (dashed black line, representing losses) and positive impacts of compensation policy designs (coloured bars, representing gains) on biodiversity (indicated by extent of native vegetation) across our four case study regions. No Net Loss (NNL) of biodiversity would have occurred if coloured bars met or exceeded dashed black lines. Coloured bars in **b**–**e** match the colour scheme shown in **a**.

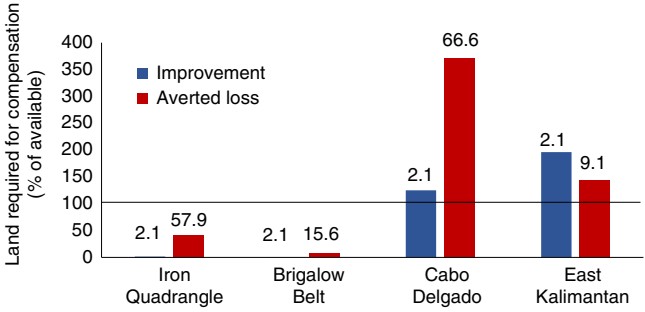

**Fig. 3 The compensation required to achieve NNL of biodiversity for our best performing Averted Loss and Improvement policy designs (see Fig. 2).** Numbers above bars show required multipliers, according to our simulated rates of biodiversity losses and gains. Columns show the required compensation area relative to the land available for compensation. In two cases (Cabo Delgado and East Kalimantan), the land required for compensation exceeds that available for protection or restoration.

the multipliers required to achieve NNL under differing local conditions (see "Best performing policies fail to achieve NNL").

The influence of local conditions (land available for compensation and counterfactual biodiversity losses and gains) on policy performance influenced some biodiversity features more than others. For example, when regulated development had large impacts on the vegetation types not threatened by unregulated development, achieving NNL was not possible with Averted Loss. Similarly, the spatial distribution of these conditions also influenced whether prioritising compensation within PAs will be more or less effective than prioritising compensation elsewhere across the landscape. Protected areas in the Brigalow Belt and Iron Quadrangle contained few opportunities to implement Improvement (i.e. only 4% and 15% was available for restoration, respectively), and their counterfactual biodiversity losses were lower, and gains higher, than those outside their protected areas. In comparison, protected areas in East Kalimantan did experience counterfactual biodiversity losses (thus prioritising Averted Loss to them could have averted some loss; Fig. 2), and 6% of areas protected in Cabo Delgado contained cleared land (thus some restoration could be undertaken; Fig. 2).

**Best performing compensation policy for biodiversity and ecosystem services.** Compensation policies increasingly seek also to address other goals related to biodiversity, such as securing the provision of ecosystem services[13,14,26] for human wellbeing[27,28]. However, rarely do they target both goals (e.g. biodiversity and ecosystem services) explicitly[29]. Instead, policies typically target one goal (e.g. biodiversity) and either assume concomitant achievement of others, or require minimisation of negative consequences, without directly exploring trade-offs. Further, no research until now has examined the extent of potential trade-offs under various policy designs and across varying local conditions. Here, we found that compensation policies performing best for biodiversity goals did not necessarily also perform well for two regulating ecosystem services: carbon storage (indicated by above ground carbon density) and sediment retention (the retention of sediment by vegetation, preventing its export to streams). We evaluated the impacts of biodiversity policy designs on these two specific ecosystem services because: biodiversity compensation likely affects them through changes in vegetation, local people in each case study region valued them[30,31], and the data and models were available to quantify their provision across all case studies (Supplementary Methods 1–4) thus enabling comparisons among them.

Impacts of development and biodiversity compensation on ecosystem services varied widely among policies, with some even achieving NNL in certain cases. In the Brigalow Belt and Iron Quadrangle, the biodiversity impacted by development was less carbon dense than the biodiversity impacted by compensation, resulting in NNL of carbon storage across all scenarios (Supplementary Fig. 2). Further, while Improvement performed better than Averted Loss for biodiversity goals, the opposite was true for carbon storage because counterfactual vegetation clearing targeted carbon-dense low-lying fertile lands[17]. We found similar results for sediment retention; some policies performed better for this goal than they did for biodiversity (e.g. Improvement in Brigalow Belt and Averted Loss in East Kalimantan), but worse for others (e.g. all policies in Iron Quadrangle and Improvement approaches in East Kalimantan; Supplementary Fig. 3).

The performance of compensation policies for ecosystem services depends on the policies' influence on biodiversity (and the local conditions explaining this, see previous section) and the links between biodiversity and ecosystem services at both the development and compensation sites[32]. Our results suggest that different compensation policies likely have widely varying impacts on ecosystem services, but that performance will also likely differ among services. Policies with multiple goals should therefore address each goal explicitly to ensure NNL is achieved for all[33]. For example, in East Kalimantan, carbon compensations perform best when they explicitly target this goal[34]. Alternatively, information on known links between goals could inform policy design. For example, if a trade-off exists between two goals, policies should seek to minimise this trade-off (e.g. through spatial prioritisation of compensation activities) and address any additional losses that it may cause[35].

**Best performing compensation policies fail to achieve conservation goals.** None of our investigated compensation policies achieved NNL of biodiversity—the biodiversity losses caused by regulated development always exceeded the gains made by compensation activities (Fig. 2). Requiring larger compensation multipliers, informed by estimates of counterfactual losses and gains, is one option to overcome such policy failure[12,20]. However, across many of our case studies, the multipliers required to achieve NNL of native vegetation were large (particularly for Averted Loss approaches, exceeding a value of 60 in Cabo Delgado for Out-of-Kind trades) compared to those often used in practice[22]. Implementing such multipliers was not possible in at least 2 of our 4 case studies (Cabo Delgado and East Kalimantan) given the limited land available for protection and restoration (Fig. 3). Other practical limitations of large multipliers may include negative trade-offs with some ecosystem services, and difficulty in protecting or restoring vegetation of a comparable condition to that removed by regulated development. Another option to achieve NNL is to prevent further development (or at least their residual biodiversity losses) once compensation options are exhausted. This could have huge implications for future development scenarios, for example this strategy of avoidance could reduce the expansion of mining in Cabo Delgado by half (Table 1). Designing policies to reflect local conditions may improve their performance, as even in situations where NNL of biodiversity and ecosystem services is theoretically achievable, many other factors may limit them in practice. Such factors may include conservation effectiveness, governance capacity and long-term monitoring and management.

Compensation is often proposed as a means to achieving regional to global-scale conservation goals. However, we found that even the best performing policies had relatively minor influence on slowing regional biodiversity loss. Across three of

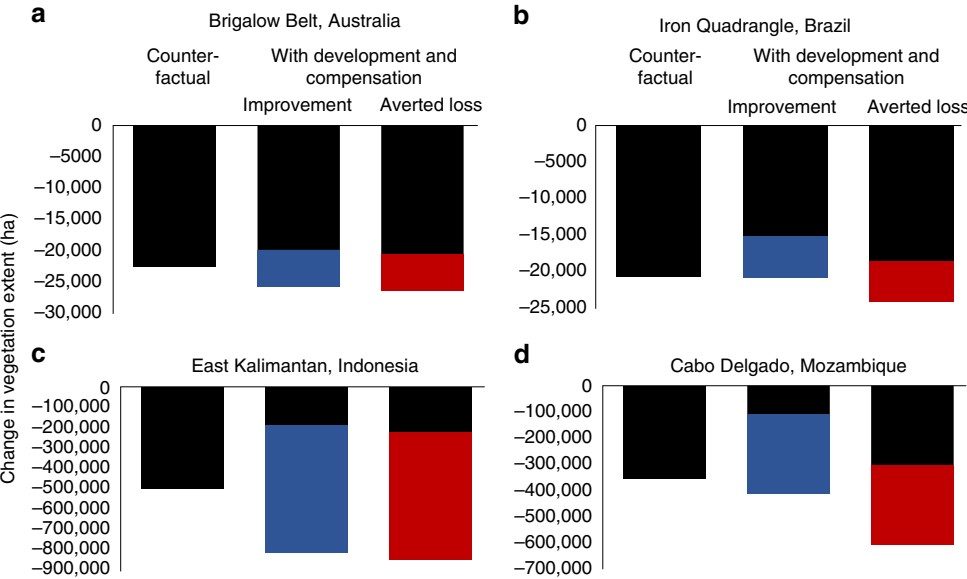

**Fig. 4 Change in native vegetation under our counterfactual scenario (without regulated development and its required compensation) and the best performing Averted Loss and Improvement policy designs (Fig. 2).** Unregulated biodiversity losses are shown in black and net biodiversity losses due to regulated development (i.e. regulated losses minus compensation gains) are shown in red (for Averted Loss) and blue (for Improvement). Panels represent four case studies: **a** Brigalow Belt, Australia; **b** Iron Quadrangle, Brazil; **c** East Kalimantan, Indonesia; **d** Cabo Delgado, Mozambique.

our four case studies, the policies we simulated reduced region-wide vegetation loss by <10%, relative to counterfactual scenarios, although in Cabo Delgado, this value reached 37%. Thus, even with perfectly implemented policies (as we assumed here), considerable losses still occur (between 3 and 13% of each region's native vegetation extent during our analysis period; Fig. 4). These losses occurred because our compensation policies, like those worldwide, were narrow in scope—they regulate only a small proportion of sectors causing biodiversity loss (Supplementary Methods 1–4). When large unregulated losses occur, compensation can only ever make a relatively small difference in overall conservation outcomes at broader spatial scales. Note: we did not assess possible biodiversity losses due to leakage or displacement of unregulated development from compensation sites to other parts of the landscape, which could further reduce the contribution that compensation makes to regional outcomes.

Our findings are consistent with the intent of many compensation policies globally, which aim to achieve NNL relative to a counterfactual scenario of without development and associated compensation[10] and, if that scenario is one of decline, maintaining decline in net terms, albeit at a lower rate, is the goal. Compensation policies with a narrow scope clearly have only limited ability to ensure that development goals do not undermine those of conservation. There are two ways to reduce this problem, in addition to applying the mitigation hierarchy to avoid and minimise biodiversity losses where possible. One is to broaden policy scope, which would force compensation approaches towards Improvement because fewer unregulated losses will exist to avert. However, expanding policy scope will increase compensation requirements, which our results suggest are already limited by land availability. A second option is to design policies to achieve biodiversity targets[36,37], rather than to maintain a counterfactual scenario of decline. This would require more challenging policy decisions about the division of responsibility for achieving conservation outcomes between industry and government, but would improve clarity on how compensation activities contribute to conservation outcomes.

**Implications for compensation policies**. Achievement of NNL is limited by two local conditions that deserve explicit consideration in compensation policy design: the extent of land available to implement compensation, and the counterfactual losses and gains within compensation sites. Our results highlight that Averted Loss approaches may outperform Improvement approaches when counterfactual biodiversity losses are high and counterfactual gains low, so long as sufficient land exists. In-Kind trades achieve greater gains for the specific vegetation types impacted by regulated development, but will perform similarly well to Out-of-Kind trades when threats are distributed evenly among types. In general, prioritising compensation near existing protected areas will achieve greater gains in regions where these sites are most threatened. However, even the best policies seemingly make a relatively small contribution towards slowing region-wide biodiversity loss. Compensation policies also have potential trade-offs with other conservation goals, such as ecosystem services. Our findings illuminate the limitations of current compensation policies; rarely do they address all significant threats to biodiversity and expanding their scope to do so would likely require more land than is available for protection or restoration. While our results suggest that policy performance may be improved by ensuring their design aligns with local conditions, expanding policy scope and requiring larger multipliers will not always be possible. This leaves impact avoidance—the first step of the mitigation hierarchy and a non-negotiable task once compensation opportunities are exhausted—key to halting biodiversity declines and achieving NNL goals.

## Methods
**Conceptual framework**. Our conceptual framework is shown in Fig. 5. Regulated development and compensation policies determine the required compensation activities, which cause biodiversity losses and gains, respectively. Here, we varied compensation policy settings, according to 18 design options illustrated in Fig. 2, and measured impacts of both development and required compensation on biodiversity (as net impacts relative to a counterfactual scenario). When biodiversity gains from compensation equalled the biodiversity losses due to development, NNL was presumed to have occurred. We also investigate the influence of interactions between policy settings and local conditions on policy performance (i.e. how close

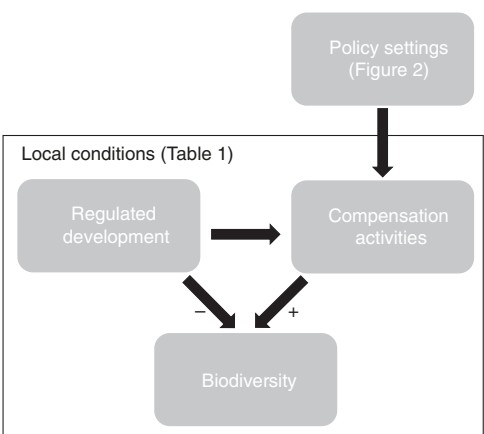

**Fig. 5 Conceptual framework used to examine the impacts of regulated development and compensation activities on biodiversity, under various policy settings and local conditions.** We manipulated compensation policy settings (see Fig. 2), measured their impact on biodiversity across four case studies that differed in local conditions (Table 1), and compared the biodiversity gains made by compensation activities to the biodiversity losses due to regulated development.

compensation activities come to achieving NNL goals) by applying our methods to four case studies that vary substantially in local conditions (Fig. 1; Table 1). Many other factors likely differ among policies (e.g. implementation and compliance) and local conditions (e.g. economic costs of acquiring land for compensation), but were held constant in our analysis.

**Modelling approach**. We developed a modelling approach (Fig. 6) to quantify the performance of 18 compensation policy designs (Fig. 2) across our four case studies.

**Step 1: quantify impacts of development on biodiversity**. We determined the extent and spatial distribution of future regulated development (i.e. development requiring compensation), which differed among case studies (Table 1 and Supplementary Information). In the Brigalow Belt and Iron Quadrangle, current compensation policies explicitly regulate mining operations (see Supplementary Methods 1 and 2). In East Kalimantan (Supplementary Methods 3) and Mozambique (Supplementary Methods 4), policies are not yet consistently applied across entire industries, thus while our definitions of regulated development are illustrative, they are also based on policy trends in these countries and the industries such policies will likely regulate in future. In Mozambique, this included mining and infrastructure developments, and in East Kalimantan included oil palm plantations and mining operations given that any primary forest cleared by this industry is regulated. Adding or removing industries from what we consider to constitute regulated development would influence our results. Simply including additional industries would require additional impacts to be compensated. Including a specific industry could also influence where compensation would be required across the landscape, particularly for In-Kind trades if industries preferentially impact particular vegetation types.

Our method for determining the footprint of future regulated development also differed among case studies. For the Iron Quadrangle and East Kalimantan, we projected future development using land-use change simulation models (see Step 3). For the Brigalow Belt and Cabo Delgado, we mapped future development using maps of known proposed projects. We overlaid these regulated development maps (either simulated or proposed, depending on the case study) with maps of native vegetation types (our indicator of biodiversity features) to quantify impacts of development in total and per vegetation type. We assumed development would completely remove vegetation wherever overlap occurred and would not be decommissioned and subsequently revegetated during the model simulation period. See Assumptions and Limitations section below for discussion of potential implications.

**Step 2: allocate compensation to the landscape**. We developed a typology of 18 policy design options currently used around the world[8,9], representing combinations of two area-based approaches to generating biodiversity gains, four ways of trading biodiversity features between development and compensation sites and three methods for prioritising compensation to the landscape (Fig. 2). Under each policy, we allocated the required amount of compensation (according to set multipliers for each approach and specified trades) and in the required configuration (according to trades and prioritisation methods) to the landscape using a model

developed in Dinamica Environment for Geoprocessing Objects (EGO)[38] that allocates compensation as new protected areas based on the constraints specified above[12]. Specifically, the model allocates compensation to previously cleared land for Improvement and to currently unprotected vegetation for Averted Loss. All compensation was allocated at the start of our simulation time period, rather than progressively over time. As such, allocation does not consider dynamic land prices or respond to diminishing land available for compensation. While, in theory, increasing demand for compensation may increase land prices and thus incentivise impact avoidance once acquisition costs exceed expected returns from development, this dynamic is rarely observed in practice. Instead, when land is scarce, compensation requirements are often eased. For example, less compensation, or other forms of compensation, such as investment in ecological research, are often permitted instead. We did not permit Improvement to restore built-up or industrialised land, given its relatively high acquisition costs and poor prospects for conversion to a natural state. Compensation could be allocated to any vegetation type for Out-of-Kind trades and to the same vegetation type for In-kind trades. Compensation was prioritised to areas of vegetation most likely to be cleared (for Averted Loss) or least likely to recover (for Improvement) for Trading up: Additional Gains, and prioritised to the vegetation types with the least of their pre-clearing extent remaining for Trading up: Rarity. Additionally, these sites were prioritised for selection either Outside protected areas, Nearby protected areas or Within protected areas, depending on the policy design combination tested. If opportunities for compensation were exhausted (i.e. insufficient unprotected vegetation exists for Averted Loss compensation), allocation of compensation ceased.

**Step 3: simulate counterfactual losses and gains**. We examined the performance of compensation policies relative to counterfactual scenarios, which we defined as the unregulated biodiversity losses and gains that would likely occur in absence of regulated development and their compensation requirements[20]. Counterfactual scenarios were simulated using land-use change models[12] developed in Dinamica EGO[38], where vegetation clearing indicated biodiversity loss and revegetation indicated biodiversity gains. Model calibration involved quantifying historic rates of land use transitions and determining their spatial determinants. We then used these calibrated models to produce spatial probability maps of land use transitions. We validated our models by simulating land use transitions during a second historic time-period and comparing simulated transitions with observed historic transitions and a null model, using fuzzy logic and an exponential decay function at multiple spatial resolutions[38]. Land use transitions, historic rates and spatial determinants, and model assumptions differed among case studies (Supplementary Methods 1–4); however, all of our calibrated models out-performed null models and were considered of sufficiently high accuracy for the purpose of this research.

To simulate counterfactual scenarios, we masked regulated development (Step 1) from current land use maps to prevent counterfactual biodiversity losses and gains occurring within these areas and used our calibrated models to simulate future land use according to historic transition rates. We simulated change at annual time steps and for at least 20 years, noting that timeframes differ among case studies (Table 2), largely due to data limitations, and mapped counterfactual losses and gains by overlaying simulated land use maps with native vegetation types. In doing so, biodiversity losses and gains were quantified over the entire simulation timeframe, rather than on a per-year basis. Previous studies investigating temporal dynamics[20] show that the benefits of Averted Loss approaches increase over time (so long as protection averts all future threats) while the benefits of Improvement approaches decrease over time as natural recovery occurs. Given that we did not examine temporal dynamics, case studies with longer time periods include more regulated development than those with shorter time periods, and correspondingly, more counterfactual biodiversity losses (since all case studies had an overall declining counterfactual). Differences in timeframes among cases do not, however, affect the multipliers required to achieve NNL (see Fig. 3), since regulated development and counterfactual biodiversity losses for each case study were quantified over the same time period. However, case studies with shorter time periods had a better chance of achieving NNL of biodiversity because less compensation was required overall and thus the risk of exceeding land available for compensation was reduced.

**Step 4: quantify compensation impacts and outcomes**. We quantified the impacts of each policy design option by examining counterfactual biodiversity losses and gains (Step 3) within areas designated for compensation (Step 2). For Averted Loss, impacts equalled the sum of counterfactual losses within compensation sites, assuming compensation would effectively avert all losses without displacing them elsewhere (i.e. without leakage; see Assumptions and Limitations section below). For Improvement, impacts equalled half the area revegetated (because we assumed a 50% success rate) minus counterfactual gains that occurred within compensation areas (also assuming no leakage). We quantified total impacts (i.e. across the case study region) on native vegetation extent and impacts per vegetation type achieved by compensation by the end of our analysis period.

We also quantified impacts of compensation on two ecosystem services by modelling spatially explicit above ground carbon storage and sediment retention for current, pre-clearing, and future landscapes. We obtained above ground carbon

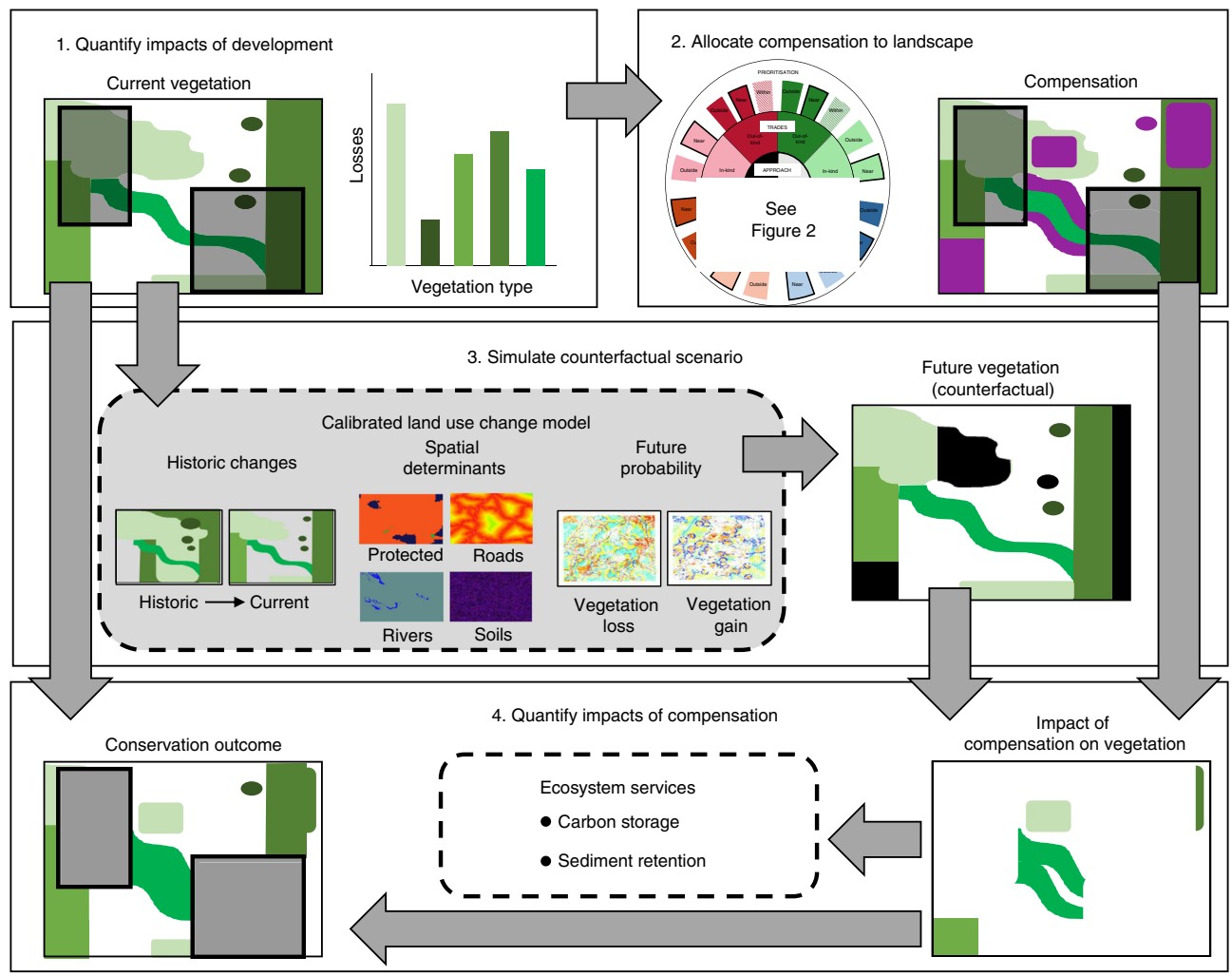

**Fig. 6 Four-step modelling approach used to investigate impacts of regulated development and compensation on biodiversity and ecosystem services.**
Step 1. Quantify impacts of regulated development (shown in shaded polygons in box) on biodiversity (here, indicated by the extent of vegetation types – green shapes in box and on graph). Step 2. Allocate compensation activities (shown as purple polygons in box) to the landscape, according to 18 policy design options. Step 3. Simulate counterfactual biodiversity losses and gains using land-use change models (black polygons indicate counterfactual losses). Step 4. Quantify impacts of compensation on biodiversity (shown as green polygons in box on right) and two ecosystem services (using spatial models), and aggregate their conservation outcomes (i.e. combined impacts of regulated development and compensation, and any counterfactual losses and gains).

**Table 2 Timeframes used to calibrate and validate our land use change models, and to simulate counterfactual biodiversity losses and gains.**

| Case study | Calibration and validation timeframe | | Counterfactual simulation timeframe | |
|---|---|---|---|---|
| Brigalow Belt | 2006, 2009, 2011 | 5 years | 2011–2020 | 9 years |
| Iron Quadrangle | 2000, 2004, 2010 | 10 years | 2010–2020 | 10 years |
| Cabo Delgado | 1992, 2004, 2015 | 23 years | 2015–2040 | 25 years |
| East Kalimantan | 1996, 2006, 2015 | 19 years | 2015–2040 | 25 years |

See Supplementary Methods 1–4 for information on individual case studies, datasets and assumptions.

storage maps from various sources (Supplementary Methods 1–4). We mapped sediment retention using the InVEST Sediment Delivery Ratio tool, a hydrologically routed RUSLE model that quantifies sediment retained by current vegetation relative to a cleared (non-vegetated) state[39,40]. Model parameters were held constant among case studies (flow accumulation: 2000 [except for in Cabo Delgado, where we used 500 due to larger spatial resolution of analysis]; Borselli's kb: 1.8; IC0: 0.5, SDR max: 0.8). We quantified impacts of regulated development and Averted Loss approaches as the difference between current and future landscapes within the compensation sites that impacted biodiversity. For Improvement, impacts equalled half the sum of differences between pre-clearing and current landscapes within compensation sites that impacted biodiversity.

Finally, we quantified case-study-wide biodiversity outcomes, as counterfactual biodiversity losses and gains, minus impacts from regulated development plus impacts of compensation activities.

**Case studies and data sources**. We applied our modelling approach to four case studies: the Brigalow Belt in Queensland, Australia; the Iron Quadrangle in Minas Gerais, Brazil; East Kalimantan in Indonesia; and Cabo Delgado in Mozambique (Fig. 1; Supplementary Figs. 5–8). Case studies differed in ways that allowed us to assess the influence of local context on compensation performance. For example, they varied in their extent of native vegetation remaining; land use and land use transitions; and compensation requirements (e.g. policy scope and extent of future

regulated development) (Table 1). Supplementary Methods 1–4 and Supplementary Tables 1–25 contain details of local context for each case study, along with specific data sources used in our analysis. While beyond the scope of our study, a theoretical treatment of local conditions, for example using a synthetic landscape and a sensitivity analysis to vary conditions and measure policy performance, could more precisely quantify interactions between policy design and local conditions and thus help provide further insight for optimal compensation design.

**Assumptions and limitations of modelling approach**. To permit analysis at broad spatial scales and across multiple case studies, our modelling approach made several simplifying assumptions, which may influence generalisation of our results.

We used the extent of native vegetation types as our proxy of biodiversity, given the spatially explicit data available across all case studies. While vegetation types may adequately indicate other levels of biodiversity (e.g. species, genetic) in some cases[41], this simplification imposes two obvious limitations. First, we were unable to consider changes in native vegetation condition or degradation because these data were not available at the extent and resolution required to conduct our analyses. Compensation policies may perform better when using condition as a proxy if, for example, development removes already degraded vegetation and compensation achieves large gains in condition by restoring degraded sites or averting loss of pristine sites. Evaluating vegetation condition may also affect the land available for compensation, given that Improvement could also restore degraded vegetation, rather than being limited to revegetating cleared land. Using native vegetation extent also overlooks biodiversity losses and gains unrelated to changes in vegetation extent or condition, such as hunting or poaching. Incorporating these threats into our analysis may increase compensation performance in some places. For example, hunting is an important driver of biodiversity loss within Cabo Delgado's protected areas and accounting for this threat would increase the performance of policies prioritising compensation Within PA. Including other proxies of biodiversity requires additional data, which does not exist at large spatial scales across all of our case studies; future research could examine the role of these factors for specific case studies at finer resolutions.

We assumed that regulated development completely removes biodiversity where it co-occurs with native vegetation. The influence of these assumptions on our results may differ among case studies, for two reasons. First, differing development types exist in each case study (i.e. mining in the Iron Quadrangle and Brigalow Belt, but also transportation corridors in Cabo Delgado and oil palm in East Kalimantan) and these may have varying impacts on biodiversity. While mining often requires land clearing for resource extraction, oil palm may house relatively more biodiversity[42,43] and thus cause lower development impacts and greater compensation performance than reported here. Second, methods used to model future regulated development differed among case studies (models were used to simulate regulated losses in the Iron Quadrangle and East Kalimantan, whereas proposed development footprints were used in the Brigalow Belt and Cabo Delgado). Since development footprints are generally larger than the vegetation actually cleared for development, our analysis may overestimate development impacts or underestimate compensation performance, in these cases and also affect the land available for compensation. However, this assumption is not expected to affect the relative impacts of development and compensation (and thus compensation performance) among policy designs.

We assumed a 50% restoration success rate and thus implemented an Improvement multiplier of 2. This assumption of restoration success relied on the restoration ecology literature, which suggests that between a third and one half of restoration projects may be successful[23,24], although sometimes success rates fall far below these rates, as is the case with seagrass restoration[44]. However, these reviews also reveal that when projects attempt to establish new habitat (as opposed to enhancing existing habitat), as is the case for compensation activities investigated in our study, and new habitats must resemble a similar condition to a reference site by a set time period, success rates are reduced even further – so our assumption is generous. While there is currently a great lack of evidence of restoration success in the offsetting context[45], recent studies suggest typically used multipliers are far from sufficient to overcome the risks of project failure[19,34]. Supplementary Figure 4 illustrates the effect of two alternative success rates (25% and 75%) on compensation impacts on biodiversity. Increasing success to 75% increased the impact of compensation and, when holding the multiplier at 2, some policy designs achieved NNL in some cases (Supplementary Fig. 4). Further research on the success of restoration (particularly informed by field studies examining real offsetting projects) is needed to better inform these assumptions. This includes an understanding of how success differs among habitat types, modes of restoration and types of initial disturbance[25]. Not only would this information permit more certainty in policy evaluations, but also to inform the multipliers required to achieve NNL, which rapidly decline as success improves.

Given the large extent of our study regions and the assumptions of our modelling methodology (Fig. 6), our results are likely influenced by several sources of uncertainty. While parameter values used by our compensation policies (i.e. multipliers and compensation success) are based on current practice, these likely vary across and among sites. Further, errors are likely evident in the many datasets used to calibrate and validate our land use change models (which were then used to simulate counterfactual biodiversity losses and gains) and ecosystem services models (used to examine trade-offs from biodiversity compensation). These data include land use maps, created by classifying satellite imagery and thus influenced by atmospheric conditions as well as errors in classification models, and spatial determinants of change (such as topography, and roads and river networks). Further, some of these sources of uncertainty may also interact with one another, causing errors to propagate throughout our analysis, particularly when simulating future counterfactual scenarios. While we could quantify some of these sources of uncertainty (indeed we discuss the influence of some of these sources in the main text), a comprehensive assessment was deemed beyond our study scope. For this reason, our results should not be used to develop specific policies in any case study, but instead used to guide high-level policy development and inform more precise evaluations in specific case studies.

**Reporting summary**. Further information on research design is available in the Nature Research Reporting Summary linked to this article.

## Data availability

Supplementary Dataset 1 contains a summary of our modelled results. GIS outputs (i.e. maps of land use change and compensation activities) are available from the corresponding author (L.J.S.) on request. All third party datasets accessed for use in this work are available via the citations provided in the main manuscript and Supplementary Information (specifically: Supplementary Table 3 for Brigalow Belt, Supplementary Table 7 for Iron Quadrangle, Supplementary Table 13 for East Kalimantan and Supplementary Table 19 for Cabo Delgado).

## Code availability

We used Dinamica EGO (version 4) to model land use change scenarios (available for free here: https://csr.ufmg.br/dinamica/) and the InVEST sediment model (version 3.2) to quantify the impact of compensation of sediment retention (available for free here: https://naturalcapitalproject.stanford.edu/software/invest).

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

## Acknowledgements

This research was supported in part by the Science for Nature and People Partnership (SNAPP) Compensatory Conservation Working Group, a partnership of The Nature Conservancy, the Wildlife Conservation Society and the National Center for Ecological Analysis and Synthesis (NCEAS) at University of California, Santa Barbara. L.J. S. acknowledges Australian Research Council Discovery Early Career Research Award (DE170100684); M.M. acknowledges Australian Research Council Future Fellowship (FT140100516). M.M. and J.S.S. were supported by The Australian Government's National Environmental Science Program through the Threatened Species Recovery Hub; and multiple co-authors received support from COMBO (COnservation, impact Mitigation and Biodiversity Offsets in Africa), which is funded by the Agence Française de Développement, the Fonds Français pour l'Environnement Mondial and the Mava foundation, among others, and implemented by the Wildlife Conservation Society, Forest Trends and Biotope. We thank Hugh Possingham and Joe Bull for helpful feedback on a previous version of the manuscript, Tom Lloyd for editing the manuscript and figures, and many others who participated in our SNAPP workshops.

## Author contributions

L.J.S., J.S.S. and M.M. conceived the ideas and designed the study; L.J.S. and J.S.S. performed analyses; L.J.S., J.S.S., J.M.K., H.M.C., K.J., K.S., M.Satar and B.S.S.F. collected secondary data; L.J.S., J.S.S., J.M.K, K.S., M.S., H.M.C., H.S.G. and A.v.H. interpreted case study results; L.J.S., J.S.S., J.E.M.W., J.P.G.J., J.M.K., H.M.C., L.B., S.E., H.S.G., V.F.G., K.J., K.S., P.P., F.Q., H.Rainer, H.Rainey, D.R., M.Satar, B.S.S.F., M.Starkey, K.t.K., R.V., A.v.H., J.A.W. and M.M. interpreted the significance and implications of case study results for compensation policies globally; L.J.S., J.S.S., J.E.M.W., J.P.G.J., J.M.K., H.M.C., L.B., S.E., H.S.G., V.F.G., K.J., K.S., P.P., F.Q., H.Rainer, H.Rainey, D.R., M.Satar, B.S.S.F., M.Starkey, K.t.K., R.V., A.v.H., J.A.W. and M.M. wrote and edited the paper.

## Competing interests

L.B. receives income from commercial contracts for consultancy services related to the development and implementation of biodiversity offset policies. All other authors declare no competing interests.
