## [Peer Review File · Nature Communications]

Reviewers' comments:

Reviewer #1 (Remarks to the Author):

This manuscript uses case studies across four world regions to examine the implications of different ecological compensation policy designs on the degree to which these programs achieve no net loss of biodiversity, carbon, and sediment retention at specific sites. The authors rely on spatial simulation models to compare scenarios of policy implementation to offset development (e.g., for mining) with 'counterfactual' no offset scenarios. They find that although such policies tend to offset some negative impacts of development, no policy scenario fully achieves no net biodiversity loss. The paper represents a substantial amount of work across a diverse author group, and attempts to be sensitive to important policy, social, ecological, and biophysical conditions across regions. Based on my literature review (I am not an expert in the literature on NNL/compensation policies) it seems that they present some novel insights into the limitations and design of these policies. Results could be influential for the field of biodiversity/ecoservice offsetting as well as for policymakers because they provide an apparently novel lens through which to view policy design and success. At the same time, I have some major concerns around the approach and presentation, as outlined below in General and Specific comments.

GENERAL COMMENTS

1. Regulated Development. It is not clear that all "regulated" development in the model is really regulated. For instance, in Indonesia, there is no explanation for why mining is included as 'regulated'. In the case of oil palm, my understanding is that ISPO has no compensation or remediation procedure (even though it is required for all producers). RSPO certification only covers around 20% of total oil palm production. Going forward, the compensation and remediation procedure only applies to companies that have recently joined the RSPO, but not to those who are already members. Instead, RSPO member producers need to avoid and/or protect forests except those that were cleared before the remediation and compensation agreement was finalized a few years ago. Thus, it is very unlikely that biodiversity loss from oil palm development would need to be fully offset in Indonesia. The justification is similarly weak for Mozambique. I would like to see a strong justification for both sites; it could be that these locations do not yet have strong remediation rules, but they serve as interesting sites to look at the hypothetical interaction between policies and local conditions.
2. Interactions between policy design and local conditions. The authors conclude that policy performance depends on local conditions, which is unsurprising. I think the more novel findings in this paper are about specific *interactions* between policy design and local conditions. The authors do a fairly good job of categorizing the policy design factors and then using these understandings to discuss policy performance. However, the local conditions are not treated with similar rigour. I would love to see a rigorous theoretical treatment of the local conditions that affect success of policies, as well as clearer discussion of how design interacts with these local conditions to determine NNL achievement outcomes. For instance, the authors imply that region size and counterfactual biodiversity loss matter, but these features and their interactions with policy design are not clearly depicted in a table or figure. Temporal factors also seem critically important, but they are not mentioned much in the manuscript.
3. Time. Model timeframes are different, and there was no apparent justification for these differences, nor a discussion about how they affect inter-site comparisons. In particular, the interaction between the number of years over which overall regional biodiversity loss is being measured and total regulated development area needs more consideration. I would like to see temporal dynamics more clearly addressed in the manuscript (potentially as part of the theoretical treatment of site specific conditions).
4. Policy recommendations. What about adding an 'optimal' conservation scenario for each site? Since the counterfactual scenario is known, you could identify the areas that would satisfy each policy for each location to achieve or get close to NNL. This would provide multipliers that would be needed to achieve NNL, at least in sites where this was physically possible. Such an addition seems like a useful exercise to inform these types of policies.
5. Locations of Compensation. The allocation of compensation sites across space and models is never clearly explained, yet for a reader, interpretation of the results depends on a clear understanding of this mechanism. Please add a clear explanation of the allocation of compensation sites across space and time.

6. Model Validation. The models are validated by comparing a null model to the calibrated model for each transition and site. The authors need to provide a justification for this choice of validation technique, as well as some discussion about the results of the validation – does outperformance of a null model have sufficiently high accuracy for the purpose of this research? Have others used this approach in the past?

7. Uncertainty. The results are not presented with uncertainty bounds, even though there are several sources of uncertainty in the inputs and models. For instance, carbon estimates typically come with uncertainty bounds – perhaps the sediment retention model also includes uncertainty? Moreover, the simulation models produce slightly different outcomes each time they are run. For a high-impact venue like nature communications, incorporating such best practices (uncertainty propagation and confidence bounds on final estimates) into the research is needed and may produce some interesting insights.

SPECIFIC COMMENTS

Main Text

L71-72. This seems a disingenuous argument. Why would heterogeneity in design and context-dependent outcomes make it hard to assess the impacts of these policies? Such variation could in contrast FACILITATE analysis because it enables research such as that presented here. I imagine the bigger barrier to understanding likely success/failure is the difficulty of developing a good counterfactual, especially when these policies are applied by governments (and are therefore wall to wall, so there are no untreated areas). This is the problem that the model solves – it generates a counterfactual which can be used in such analysis. I think that this paragraph should be re-written to either more clearly state why variation in policies and local conditions complicate analysis, or to make a different argument (which may be along the lines that no research has yet leveraged this diversity in an empirical setting to assess how policy design interacts with local conditions to affect outcomes).

L79-82. Great. Have these factors (the influence of policy design and local conditions) been explored by previous empirical research? It seems that the concepts presented in this paper are theoretically quite clear (e.g., it is not surprising that counterfactual biodiversity loss interacts with the multiplier to determine whether the policy meets its goals) but I cannot tell from the introduction whether such interactions have been highlighted – or not – in prior empirical or theoretical research. It would be helpful to clearly reference past work or lack thereof in such statements so that that the knowledge gap is clear.

L85-86. Why were these cases chosen – are they particularly good places to do this work, and if so, why? Are there compensation mechanisms already acting in these regions? Some justification for site selection should be provided in the main text.

L118-119. This argument is not clear – please explain why expanding policy scope would require more land than is available.

Figure 2. The bar graphs in this figure would be much easier to read if the bars were labelled with the policy design. Otherwise the reader must remember, or constantly refer to, the key above the bar graphs. Same comment for supplement figures.

L129-130. What is special about 'out of kind' trades that makes them suitable to PAs?

L146-148. Perhaps I misunderstand but it seems that an averted loss multiplier of 4 would compensate for a counterfactual biodiversity loss of 25% ($.25 \times 4 = 1$).

L153. Why is a 50% re-vegetation success assumption optimistic? Justification needed to explain logic around this statement.

L196. This is not a very surprising result. Of course policy performance depends on local conditions!

L201-204. It is important to mention that this will always happen over long time frames in models that allow regulated development to increase each year.

L251-252 "because local people in each case study region valued them" - This statement requires a citation or data that back up the claim for each region, or additional detail about why people value these services. I agree these ecoservices are interesting because they have local/regional (sediment) and global (carbon) relevance. The sediment issue is likely important to some local residents, but in many cases carbon sequestration is of greater interest to governments and may not be of relevance to local residents.

L273-275. Good paper exploring this concept: O'Connell, Christine S., et al. "Balancing tradeoffs: Reconciling multiple environmental goals when ecosystem services vary regionally." *Environmental Research Letters* 13.6 (2018): 064008.

L299 – More interesting question, perhaps: What would be the overall impact of 'optimal' policies in each case? How good can it get given that there are other causes of biodiversity loss in these regions?

L300-302. Another hypothetical reason for this type of result could be that the major driver of biodiversity loss in a region is the regulated development and there is simply no potential for avoided loss or improvement outside of the development footprint. That does not seem to be the case in any of your case studies, but perhaps worth mentioning.

L315-316. Not sure what this means. How can development and conservation goals be consistent? Figure 4. I don't understand this graph. Are the colors the net positive land conservation under the scenario? If so, why do they offset more than the counterfactual vegetation change?

L385-395. It is not clear how these are allocated to the landscape. A more detailed explanation of this part of the model is needed.

L408. Please explain why regulated development was masked.

L457-458. What is meant by "while vegetation types may adequately indicate other levels of biodiversity in some cases"? I guess 'other' refers to 'other than vegetation' but this is not clear.

L459. Why were the authors 'unable' to consider changes in native vegetation condition or degradation? Were the datasets unavailable?

L478. A better citation is the new IUCN report: Meijaard, E., et al. "Oil palm and biodiversity: A situation analysis by the IUCN Oil Palm Task Force." (2018): 116p.

Supplement

L41. Why were transition rates from the 2006 to 2009 period used (rather than those from 2009-2011)?

L47. Why were above and belowground carbon included in the Australia case, but not the others?

L143-146. Needs citations. Also, only a proportion of oil palm is RSPO certified...and future developments are unlikely to be part of the compensation and remediation mechanism...so the coverage of this mechanism in space is likely limited. What regulations require compensation for mining operations?

Table S13 – why is shrub to mining 'regulated loss' given that shrub is not classified as native vegetation?

L204-205. Why were these projects considered as regulated development?

Reviewer #2 (Remarks to the Author):

Comments about the introduction section:

In order to be clear about what is compared between the different sites a specific theoretical framework regarding policy design should be adopted such as the institutional analysis and diagnosis (IAD) framework of Ostrom.

Comments about the result and discussion section:

My main concern about the paper is that even if the modeling process seems scientifically sound (but not completely understandable), the results appear as depending mainly on arbitrary choices carried out by the authors regarding several parameters of the model, especially the multipliers (2 for improvement and 4 for averted), the native biodiversity state (pristine), the rate of success of the restoration works (50%).

What is really difficult to understand for me is why these parameters have not been calibrated from data collected in the field. Indeed, we could expect to have an empirical analysis based on field work and not only a modelling work for this type of article.

At the end I even did not really understand why it was necessary to have « real » case studies in this paper, except to provide basic information on specific development projects. In addition no sensitivity analysis has been carried out to assess how the results depend on these arbitrary assumptions.

Unfortunately it leads to have both scientific limitations of a broad but rough comparative analysis using big database (lack of qualitative analysis) and of an in-depth fieldwork based on few study sites (lack of quantitative comparisons)... without the strengths of these methods.

I would have suggested to start from a big database regarding this topic (for example the IUCN

database regarding the compensation policies around the world) and to create some generic categories of countries having common policy designs (using the 18 compensation policy designs options or others). The IUCN database is mentioned (once) but not used at all, which seems surprising. With such an approach, it would have been possible to draw a broad picture of the different level of compliance regarding compensation around the world and to go further in the detail next.

Figure 2 (which summaries the main results of the paper) says three simple things for someone interested in improving ecological compensation methods. First, the no net loss is not reached even in the best policy configuration. Second, it is necessary to have an « improvement approach » since the ecological benefits are always higher than for the « averted loss ». Third, there is no difference between trading types and prioritisation types if you adopt an « improvement approach ».

The first result is not surprising since the "arbitrary choices" adopted for calibrating the models are all based on the idea that the compensation does not work. For example, why assuming that the success rate of « improvement » is 50%, without mentioning any source for supporting this assumption ? It is necessary to have, at least, a literature review mentioning this rate. Another option could have been to mention previous results (regarding this rate) in the study sites. Another "arbitrary choice" is that the impacted native vegetation was in its pristine state. With the adoption of such standards, it is not surprising that one of the conclusion of the paper was that the compensation measures don't allow to reach a no net loss goal. Again the main problem is not that these assumption are wrong or right, but that no fieldworks have been carried out to check them (even with a very rapid assessment method).

The second result highlighted by Figure 2 (improvement better than averting) is so obvious that it seems not necessary to have a modelling process like this one to say that.

The third result lead to consider that all this modelling process does not allow to disentangle the parameters to target for improving the success of offsetting policies (only in Cabo Delgado it is possible to highlight that « in kind » approach is worse than others). In conclusion these results cannot be considered as very useful for managers.

It was also not clear for me why it is necessary to assess the impact of compensation on two ecosystem services, neither why it was these two ecosystem services which have been chosen, and what is the added value of this information for this paper. I understood that it helps to question the trade-offs but since the starting point of the paper is the "regulated development" (with no legal requirements for the loss of ecosystem services) the ecosystem service section appears a little bit as off-topic.

In summary I am sorry to write that the results are not very convincing, neither very exciting.

Comments about the conclusion section:

One of the core message of the paper is that the NNL goal is difficult to reach because there are not enough land available. Unfortunately it is not obvious, for the reader, to understand how the availability of the land is defined in the paper. I guess that it was the surface of vegetation in bad condition and the surface of the cleared land which allow to defined it. If it is the definition retained it can be highlight that other types of landuse could be eligible for compensation, even some built areas. What is sure it that a core issue is not mentioned in the paper regarding the availability of the lands. Indeed, the availability of land is at a given price. Higher is the price you are willing to pay for buying a land which will be used for compensation higher are the available lands for compensation. Indeed, the problem of land availability is, at a certain extent, only a problem of price of the lands. If the price of the compensation is very high (as in certain American states) it is possible to buy some lands where there are buildings for examples, and restore these lands with a great ecological benefit.

Another limit of the conclusion is that there are no discussion regarding other works carried out on this topic in order to compare the results and discuss/interpret potential differences.

Comments about the materials and methods section:

The step 1 is OK for me.

The step 2 is far to be so convincing for me. First the 18 policy design options are assumed to be « currently used around the world ». OK but it is necessary to give some references to feed this assumption. Next, the reader would like to know a little bit more about the « required amount of compensation » and the « required configuration » but also the « model developed in Dinamica EGO ». In fact all the sentence gives the feeling to the reader that the analysis is based on a « black box » impossible to understand if you are not involved in the work.

I don't understand the step 3. The first sentence of the paragraph is « we quantified impacts of compensation (see Step 4) relative to... ». It is a little bit confusing for the reader to try to understand the step 3 starting with the notification that it is necessary to read the step 4 before (assuming that the step 4 is after the step 3...). The following sentences are not more clear for me. I am sorry to write that after three readings I was still completely lost. I was unable to understand, at the end, what was concretely the « counterfactual biodiversity losses and gains ». So the step 4 was not easier to understand. Maybe it is because I am not enough familiar with this type of models. But I suspect it is very challenging to understand this paragraph even if you are familiar with this type of model.

Comments about the assumptions and limitation section

What is very frustrating in the limitations mentioned by the authors is that some of them could have been fixed with a two weeks field works.

Clearly, it can be expected from this kind of transversal analysis, based on only 4 local case studies, that a real fieldwork was carried out in each of these sites.

Reviewer #3 (Remarks to the Author):

This study takes up the increasingly prominent notion of "no net loss" conservation policies and uses a modeling approach to test whether and which policy options and different local conditions come close to achieving "no net loss" of biodiversity outcomes. Interestingly, not a singly policy option in not one of the four study areas in different continents seems able to fulfill the "no net loss" promise.

The research question addressed is very timely, the methodology seems robust, and the insights are interesting and relevant to a broad range of target audiences. Overall the paper is well-written and structured. But as it contains a lot of content and many potentially valuable insights, I'd urge the authors to make the paper as clear and understandable as possible.

In particular - while I find the results and discussion on policy design clear - I am struggling a bit with the part on local conditions (availability of land and counterfactual biodiversity losses and gains). Here, I feel the reader needs more explanation. How was the amount of available land for compensation defined? Within which geographic boundaries? A clearer definition with examples of counterfactual losses and gains may also be helpful.

As for the policy design - what I read from Figure 2 is that "Improvement" options generally perform much better than "Averted Loss" options. And that within the "Improvement" category, most trade and prioritisation options perform similarly in the four study areas. Shouldn't a stronger policy message be taken out of that?

The additional consideration of two ecosystem services adds value to the manuscript. While it is intuitive to study additional effects on carbon storage, the consideration of sediment retention appears a bit random. Can you find some stronger arguments of why outcomes of compensation in terms of sediment retention should be considered? Is that really an ecosystem service of outstanding importance in the four study areas / for the types of development studied?

The study yields many results that are relevant for national and global conservation policies. I'd like to encourage the authors to distill some stronger policy insights, ideally in the conclusions section. For example, the authors mention twice that further development may have to be prevented once compensation options are exhausted. This could become a strong policy message.

Also, shouldn't your results give reason to question the usefulness of the term "no net loss" of biodiversity - given that not a singly of your options achieved no net loss? Is NNL no more than a buzzword then?

Minor remarks:

L. 128-130: I have difficulties understanding the justification in the note. Consider rewording / clarifying.

L. 200-201: Something seems missing in this sentence.

Reviewer #1 (Remarks to the Author):

This manuscript uses case studies across four world regions to examine the implications of different ecological compensation policy designs on the degree to which these programs achieve no net loss of biodiversity, carbon, and sediment retention at specific sites. The authors rely on spatial simulation models to compare scenarios of policy implementation to offset development (e.g., for mining) with 'counterfactual' no offset scenarios. They find that although such policies tend to offset some negative impacts of development, no policy scenario fully achieves no net biodiversity loss. The paper represents a substantial amount of work across a diverse author group, and attempts to be sensitive to important policy, social, ecological, and biophysical conditions across regions. Based on my literature review (I am not an expert in the literature on NNL/compensation policies) it seems that they present some novel insights into the limitations and design of these policies. Results could be influential for the field of biodiversity/ecoservice offsetting as well as for policymakers because they provide an apparently novel lens through which to view policy design and success. At the same time, I have some major concerns around the approach and presentation, as outlined below in General and Specific comments.

RESPONSE: We thank R1 for their comments and constructive feedback on our manuscript. Below we respond to each point raised, illustrating how they have been addressed by our revisions. Please note that page and line numbers quoted in our responses refer to the tracked changes version of our revised manuscript.

GENERAL COMMENTS

1. Regulated Development. It is not clear that all "regulated" development in the model is really regulated. For instance, in Indonesia, there is no explanation for why mining is included as 'regulated'. In the case of oil palm, my understanding is that ISPO has no compensation or remediation procedure (even though it is required for all producers). RSPO certification only covers around 20% of total oil palm production. Going forward, the compensation and remediation procedure only applies to companies that have recently joined the RSPO, but not to those who are already members. Instead, RSPO member producers need to avoid and/or protect forests except those that were cleared before the remediation and compensation agreement was finalized a few years ago. Thus, it is very unlikely that biodiversity loss from oil palm development would need to be fully offset in Indonesia. The justification is similarly weak for Mozambique. I would like to see a strong justification for both sites; it could be that these locations do not yet have strong remediation rules, but they serve as interesting sites to look at the hypothetical interaction between policies and local conditions.

RESPONSE: R1 is correct – while some form of compensation policy currently exists in each case study, those in Mozambique and East Kalimantan are not yet consistently applied across entire sectors. For these cases, our definition of regulated development is necessarily illustrative, but also based on policy trends (informed by in-country experts) and the industries these policies will likely regulate in future. In the case of Kalimantan, the basis for the offset criteria were from the voluntary RSPO process (<https://rspo.org/certification>) which the Indonesian government is considering as a model for broader expansion, and so exploring *ex-ante* the implications of different approaches to doing so (as we do here) is, we think, particularly valuable. In the case of Mozambique, it passed a new regulation in 2017 (Regulation 89/2017 of 21 December of the Law on the Protection, Conservation and Sustainable Development of Biological Diversity) which requires No Net Loss of biodiversity, for which specific guidance is now being explored – and our results already are feeding directly into this process. We have clarified our justifications in our Introduction (P3 L107–109) and added a full explanation to our Methods (P12–13 L456–467; see quoted text below), including a

sentence about how adding or removing regulated industries from case studies will likely affect our results (P12–13 L466–470).

“In the Brigalow Belt and Iron Quadrangle, current compensation policies explicitly regulate mining operations (see Supporting Information A and B). In Mozambique and East Kalimantan, policies are currently under development and entire industries not yet consistently captured by compensation policies, so the definitions of regulated development we used in our case studies were based on policy trends in these countries and the industries such policies will likely apply to in future. In Mozambique, this included mining and infrastructure developments, and in East Kalimantan included oil palm plantations and mining operations given that any primary forest cleared by these industries is a regulated impact. Adding or removing industries from what we consider to constitute regulated development would influence our results. For example, including additional industries would require additional impacts to be compensated. Including a specific industry could also influence where compensation would be required across the landscape, particularly for In-Kind trades if industries disproportionately impact particular vegetation types.”

2. Interactions between policy design and local conditions. The authors conclude that policy performance depends on local conditions, which is unsurprising. I think the more novel findings in this paper are about specific *interactions* between policy design and local conditions. The authors do a fairly good job of categorizing the policy design factors and then using these understandings to discuss policy performance. However, the local conditions are not treated with similar rigour. I would love to see a rigorous theoretical treatment of the local conditions that affect success of policies, as well as clearer discussion of how design interacts with these local conditions to determine NNL achievement outcomes. For instance, the authors imply that region size and counterfactual biodiversity loss matter, but these features and their interactions with policy design are not clearly depicted in a table or figure. Temporal factors also seem critically important, but they are not mentioned much in the manuscript.

RESPONSE: We agree with R1 and thank them for highlighting this improved framing of our analysis. We have edited our manuscript to state that our study examines interactions between policy design and the local conditions represented by our set of varied case studies (P2 L92; P7 L234–235). We have also edited our manuscript title accordingly (P1 L3). To improve our treatment of local conditions, we have summarised differences among case studies in a new Table (Table 1), which illustrates how variation in their size, remaining biodiversity extent, counterfactual biodiversity losses and gains, and land available for compensation affects the performance of different policy design elements (P4 L129–135). We now refer specifically to these differences in our Discussion section (see section “Policy performance also depends on local conditions”; P7 L232). We have also stated that a more theoretical examination of local conditions is a necessary next step in designing effective policies and lay out an analysis required to achieve this (P15 L581–585), and added that, although exploring temporal dynamics was also beyond our scope, other studies demonstrate that benefits of Averted Loss approaches increase over time (so long as protection averts all future threats), while those of Improvement approaches decrease over time in regions where natural biodiversity recovery occurs (P14 L526–530).

3. Time. Model timeframes are different, and there was no apparent justification for these differences, nor a discussion about how they affect inter-site comparisons. In particular, the interaction between the number of years over which overall regional biodiversity loss is being measured and total regulated development area needs more consideration. I would like to see temporal dynamics more clearly addressed in the manuscript (potentially as part of the theoretical treatment of site specific conditions).

RESPONSE: Simulation timeframes differ among case studies (as shown in our new Table 2) because of differences in available data (P14 L521–522). The Brigalow Belt and Iron Quadrangle case studies were calibrated and validated using 10 and 5 years of data, respectively, and thus simulations were run for shorter time periods (2010 to 2020; 10 years). Cabo Delgado and East Kalimantan case studies were calibrated and validated using 23 and 19 years of data, thus simulations could reasonably be run for longer time periods (2015–2040; 25 years). We have also discussed the implications of differing simulation timeframes (P14 L527–538; see quoted text below).

“Given that we did not examine temporal dynamics, case studies with longer time period include more regulated development than those with shorter time periods, and correspondingly, more counterfactual biodiversity losses (since all case studies had an overall declining counterfactual). Differences in timeframes among cases do not, however, affect the multipliers required to achieve NNL (see Figure 3), since regulated development and counterfactual biodiversity losses for each case study were quantified over the same time period. However, case studies with shorter time periods had a better chance of achieving NNL of biodiversity because less compensation was required overall and thus the risk of exceeding land available for compensation is reduced.”

4. Policy recommendations. What about adding an ‘optimal’ conservation scenario for each site? Since the counterfactual scenario is known, you could identify the areas that would satisfy each policy for each location to achieve or get close to NNL. This would provide multipliers that would be needed to achieve NNL, at least in sites where this was physically possible. Such an addition seems like a useful exercise to inform these types of policies.

RESPONSE: We have further highlighted the multipliers that would be required to achieve NNL in the different case study sites (previously shown in Figure 3 and referred to on P6 L191, now also on P9 L339). We have now also clarified the situations under which these could be considered “optimal” – i.e. when seeking to achieve NNL of biodiversity, without regards to the type of vegetation compensated, its condition or trade-offs with ecosystem services (P9 L342–344). We also identify an alternative ‘optimal’ scenario, where all biodiversity losses due to regulated development are avoided once compensation opportunities are exhausted (P9 L346–349).

5. Locations of Compensation. The allocation of compensation sites across space and models is never clearly explained, yet for a reader, interpretation of the results depends on a clear understanding of this mechanism. Please add a clear explanation of the allocation of compensation sites across space and time.

RESPONSE: We have included a more detailed, revised explanation of how our model allocates compensation to the landscape (P13 L491–502; see quoted text below).

“Specifically, the model allocates compensation to cleared land for Improvement and to unprotected vegetation for Averted Loss. We did not permit Improvement to restore built-up or industrialised land, given its relatively high acquisition costs and poor prospects for conversion to a natural state. Compensation could be allocated to any vegetation type for Out-of-Kind trades and to the same vegetation type for In-kind trades. Compensation was allocated to areas of vegetation most likely to be cleared (for Averted Loss) or least likely to recover (for Improvement) for Trading up: Additional Gains, and to the vegetation types with the least of their preclearing extent remaining for Trading up: Rarity. Additionally, these sites are prioritised for selection either Outside protected areas, Nearby protected areas or Within protected areas, depending on the policy design combination tested. If opportunities for compensation are exhausted (i.e. insufficient unprotected vegetation exists for Averted Loss compensation), allocation of compensation ceases.”

6. Model Validation. The models are validated by comparing a null model to the calibrated model for each transition and site. The authors need to provide a justification for this choice of validation technique, as well as some discussion about the results of the validation – does outperformance of a null model have sufficiently high accuracy for the purpose of this research? Have others used this approach in the past?

RESPONSE: We have clarified the procedure used to validate our land use change models, indicating that this is standard practice when using Dinamica EGO and providing references to support the assertion that our model performance is sufficient (P13 L506–519). However, we also now mention this source of uncertainty in our limitation section (P16 L638–652; see response to comment #7).

7. Uncertainty. The results are not presented with uncertainty bounds, even though there are several sources of uncertainty in the inputs and models. For instance, carbon estimates typically come with uncertainty bounds – perhaps the sediment retention model also includes uncertainty? Moreover, the simulation models produce slightly different outcomes each time they are run. For a high-impact venue like nature communications, incorporating such best practices (uncertainty propagation and confidence bounds on final estimates) into the research is needed and may produce some interesting insights.

RESPONSE: The reviewer is quite right that our methodology likely involves many sources of uncertainty. We could quantify some of their effects on our results, for example, re-running all land use change models given the errors in datasets used to calibrate them (when such estimates are provided) and the validation results we have produced, conduct a Monte Carlo analysis to determine the effect of model stochasticity and a sensitivity analysis to determine the influence of chosen parameters such as revegetation success rates, and recalculate carbon storage results using uncertainty estimates to bound upper and lower limits. However, this still would not comprehensively quantify all sources of uncertainty and their interactions. For this reason, we instead add text to the Methods discussing likely sources of uncertainties and reiterate that our results should not be used to develop specific policies in any case study regions, given that many of our model parameters are based on assumptions and hypothetical future scenarios, but instead be used to guide high-level policy development principles and inform more precise model development in specific case studies (P16 L638–652).

SPECIFIC COMMENTS

Main Text

8. L71-72. This seems a disingenuous argument. Why would heterogeneity in design and context-dependent outcomes make it hard to assess the impacts of these policies? Such variation could in contrast FACILITATE analysis because it enables research such as that presented here. I imagine the bigger barrier to understanding likely success/failure is the difficulty of developing a good counterfactual, especially when these policies are applied by governments (and are therefore wall to wall, so there are no untreated areas). This is the problem that the model solves – it generates a counterfactual which can be used in such analysis. I think that this paragraph should be re-written to either more clearly state why variation in policies and local conditions complicate analysis, or to make a different argument (which may be along the lines that no research has yet leveraged this diversity in an empirical setting to assess how policy design interacts with local conditions to affect outcomes).

RESPONSE: We have clarified that variation among policies and projects makes these analyses difficult because it is not clear which differences explain relative outcomes (P2 L77–78). We have

also added R2's point about the challenge of developing good counterfactuals in quantifying the benefits of compensation activities (P2 L85–87), and identified our research gap as the interactions between policy settings and local conditions (P3 L92).

9. L79-82. Great. Have these factors (the influence of policy design and local conditions) been explored by previous empirical research? It seems that the concepts presented in this paper are theoretically quite clear (e.g., it is not surprising that counterfactual biodiversity loss interacts with the multiplier to determine whether the policy meets its goals) but I cannot tell from the introduction whether such interactions have been highlighted – or not – in prior empirical or theoretical research. It would be helpful to clearly reference past work or lack thereof in such statements so that that the knowledge gap is clear.

RESPONSE: We now clearly state that while previous research has explored the effectiveness of specific compensation policies in various case study locations, ours is the first to systematically examine a wide range of policy settings across a varied set of case studies (P3 L93–95).

10. L85-86. Why were these cases chosen – are they particularly good places to do this work, and if so, why? Are there compensation mechanisms already acting in these regions? Some justification for site selection should be provided in the main text.

RESPONSE: Our case studies represent a diverse set of local conditions (now summarized in Table 1), which permit our analysis of how policy performance varies among contexts (P3 L94–96). We have also clarified that, while all cases currently have some form of compensation policy, those in Mozambique and East Kalimantan are still under development and thus our treatment of these policies was more hypothetical and exploratory than the other two (P3 L107–108; P12 L459–466).

11. L118-119. This argument is not clear – please explain why expanding policy scope would require more land than is available.

RESPONSE: We have reworded this argument for clarity: *“Further, expanding policy scope to address these unregulated sectors would, in many cases, require more land than would be available for compensation under our simulated development scenarios”* (P4 L147–149).

12. Figure 2. The bar graphs in this figure would be much easier to read if the bars were labelled with the policy design. Otherwise the reader must remember, or constantly refer to, the key above the bar graphs. Same comment for supplement figures.

RESPONSE: We felt that adding the full description of each policy design option (e.g. “Averted Loss, Out-of-Kind, Outside PAs”) underneath each column of Figure 2 would clutter the figure with text. We could instead label bars with a code for each policy option, but the reader would still need to refer to the key for these codes. Instead we have just included one component of the policy design – the approach – on the x axis of Figure 2.

13. L129-130. What is special about ‘out of kind’ trades that makes them suitable to PAs?

RESPONSE: We have clarified that both in kind and ‘out-of-kind’ trades could hypothetically be used Within PAs; however, we found that very few opportunities for ‘out-of-kind’ trades within protected areas, because protected areas contained limited sites of cleared land for Improvement approaches, and experienced low rates of counterfactual losses needed to achieve impacts with Averted Loss approaches. Because of this, we did not investigate the even more restrictive option of ‘in-kind’ trades (P5 L158–161).

14. L146-148. Perhaps I misunderstand but it seems that an averted loss multiplier of 4 would compensate for a counterfactual biodiversity loss of 25% (.25*4 = 1?).

RESPONSE: The reason that 20% is the correct figure is because the multiplier acts as a ratio – not a percentage. So, if 4 units are conserved for every 1 unit lost, a total of 20% of the original vegetation is lost (1 unit of 5) and 80% remains (4 units of 5). This is an outcome equivalent to a counterfactual loss of 20%.

15. L153. Why is a 50% re-vegetation success assumption optimistic? Justification needed to explain logic around this statement.

RESPONSE: We have now justified our assumed 50% revegetation success rate, using systematic reviews published in the restoration ecology literature (P6 L180–181) and expanded on this ‘optimistic’ assumption in the Methods (P16 L626–636; see quoted text below).

“We assumed a 50% restoration success rate and thus implemented an Improvement multiplier of 2. This assumption of restoration success relied on the restoration ecology literature, which suggests that between a third and one half of restoration projects are successful^{23,24}. However, these reviews also reveal that when projects attempt to establish new habitat (as opposed to enhancing existing habitat), as is the case for compensation activities investigated in our study, and new habitats must resemble a similar condition to a reference site by a set time period, success rates are reduced even further – so our assumption is generous. While there is currently a great lack of evidence of restoration success in the offsetting context⁴³, recent studies suggest typically used multipliers are far from sufficient to overcome the risks of project failure^{19,33}. Further research on the success of restoration (particularly informed by field studies examining real offsetting projects) is needed to better inform these assumptions.”

16. L196. This is not a very surprising result. Of course policy performance depends on local conditions!

RESPONSE: We now discuss how policy performance depends on interactions between policy design and local conditions (P3 L92; P8 L234–236) and have reworded our title accordingly (P1 L3).

17. L201-204. It is important to mention that this will always happen over long time frames in models that allow regulated development to increase each year.

RESPONSE: We have added this point to clarify that *“there was insufficient unprotected vegetation to protect for Averted Loss, or cleared land to restore for Improvement, to allow compensation for all regulated development simulated over our investigated timeframe”* (P7 L236–238). We have also clarified the effects of our simulated time periods on expected regulated development, its compensation requirements, remaining land availability, and achievement of>NNL of biodiversity (P14 L524–538).

18. L251-252 “because local people in each case study region valued them” - This statement requires a citation or data that back up the claim for each region, or additional detail about why people value these services. I agree these ecoservices are interesting because they have local/regional (sediment) and global (carbon) relevance. The sediment issue is likely important to some local residents, but in many cases carbon sequestration is of greater interest to governments and may not be of relevance to local residents.

RESPONSE: We have added citations to support our claim that people in each case study value the ecosystem services of sediment retention (for its water quality benefits) and carbon storage (for mitigating climate change) (P8 L304–308).

19. L273-275. Good paper exploring this concept: O’Connell, Christine S., et al. "Balancing tradeoffs: Reconciling multiple environmental goals when ecosystem services vary regionally." *Environmental Research Letters* 13.6 (2018): 064008.

RESPONSE: Thank you for this recommendation – we have included this reference in our manuscript to support the following sentence: *“For example, if a trade-off exists between two goals, policies should seek to minimise this trade-off (e.g. through spatial prioritisation of compensation activities) and address any additional losses that it may cause (O’Connell et al. 2018)”* (P9 L328–330).

20. L299 – More interesting question, perhaps: What would be the overall impact of ‘optimal’ policies in each case? How good can it get given that there are other causes of biodiversity loss in these regions?

RESPONSE: We have included more discussion around the multipliers required to achieve NNL of biodiversity in each case study (P9 L337–342) and reiterated that, when these requirements exceed the land available for compensation activities, regulated development activities must be avoided if NNL is to be achieved (P9 L345–349; P11 L412–414).

21. L300-302. Another hypothetical reason for this type of result could be that the major driver of biodiversity loss in a region is the regulated development and there is simply no potential for avoided loss or improvement outside of the development footprint. That does not seem to be the case in any of your case studies, but perhaps worth mentioning.

RESPONSE: We agree that this absolutely could occur for Averted Loss approaches and, although none of the policies explored in our analysis were so broad in scope that there were not sufficient unregulated losses to avert, our manuscript does suggest that one consequence of broadening policy scope is that it would force compensation approaches towards Improvement because fewer unregulated losses would exist to avert (P10 L374–375).

22. L315-316. Not sure what this means. How can development and conservation goals be consistent?

RESPONSE: We have edited this sentence for clarity: *“Compensation policies with a narrow scope clearly have only limited ability to ensure that development goals do not undermine those of conservation.”* (P10 L372).

23. Figure 4. I don’t understand this graph. Are the colors the net positive land conservation under the scenario? If so, why do they offset more than the counterfactual vegetation change?

RESPONSE: We have edited Figure 4 legend to clarify that the counterfactual bar represents unregulated biodiversity losses (P10 L385–386). The black portion of bars “with development and compensation” thus show reduced unregulated losses but also additional loss from regulated sectors.

24. L385-395. It is not clear how these are allocated to the landscape. A more detailed explanation of this part of the model is needed.

RESPONSE: We have edited this paragraph to explain how Dinamica EGO allocates compensation according to the constraints of each policy design (P13 L489–499; see also response to comment #5). We have also added a supporting reference when the modelling environment is first mentioned.

25. L408. Please explain why regulated development was masked.

RESPONSE: We have explained that regulated development was masked from land use maps to prevent (unregulated) counterfactual biodiversity losses and gains within these areas (P14 L521).

26. L457-458. What is meant by “while vegetation types may adequately indicate other levels of biodiversity in some cases”? I guess ‘other’ refers to ‘other than vegetation’ but this is not clear.

RESPONSE: We have clarified that “other” refers to other levels of biodiversity (e.g. species, genetic) (P15 L594).

27. L459. Why were the authors ‘unable’ to consider changes in native vegetation condition or degradation? Were the datasets unavailable?

RESPONSE: We have clarified that datasets on vegetation condition were unavailable at the extent and resolution required to conduct our analyses (P15 L595–596).

28. L478. A better citation is the new IUCN report: Meijaard, E., et al. "Oil palm and biodiversity: A situation analysis by the IUCN Oil Palm Task Force." (2018): 116p.

RESPONSE: Thank you – we have added this reference to our manuscript (P15 L615).

Supplement

29. L41. Why were transition rates from the 2006 to 2009 period used (rather than those from 2009-2011)?

RESPONSE: We clarified that “*regrowth and clearing rates of Brigalow were unusually high during this timeframe¹¹ and so we chose to use the more conservative rate*” (SI P3 L42–44); however, we also note that “*using higher rates of clearing would result in greater performance of Averted Loss approaches¹⁰*” (SI P3 L44–45).

30. L47. Why were above and belowground carbon included in the Australia case, but not the others?

RESPONSE: While the VAST model calculates carbon density in above and below ground carbon pools, our analysis uses only the above ground carbon density for all cases (SI P3 L50–51).

Reviewer #2 (Remarks to the Author):

RESPONSE: We thank R2 for their comments and constructive feedback on our manuscript. Below we respond to each point raised, illustrating how their concerns have been addressed by our revisions. Please note that page and line numbers quoted in our responses refer to the tracked changes version of our revised manuscript.

Comments about the introduction section:

31. In order to be clear about what is compared between the different sites a specific theoretical framework regarding policy design should be adopted such as the institutional analysis and diagnosis (IAD) framework of Ostrom.

RESPONSE: Thank you for this suggestion. While the Ostrom's IAD framework is useful when examining holistic outcomes of policy designs across social-ecological systems, our analysis focuses solely on differences in the technical settings of policies and holds constant socio-ecological dynamics. To address R2's comment, we have added a new figure illustrating our theoretical framework (Figure 5). This framework is mentioned in the Introduction (P3 L113–114) and fully described in the Methods to illustrate the variables being manipulated and measured within, and compared among, case studies (P11 L417–433).

Comments about the result and discussion section:

32. My main concern about the paper is that even if the modelling process seems scientifically sound (but not completely understandable), the results appear as depending mainly on arbitrary choices carried out by the authors regarding several parameters of the model, especially the multipliers (2 for improvement and 4 for averted), the native biodiversity state (pristine), the rate of success of the restoration works (50%).

RESPONSE: We have further justified our model parameters. Compensation multipliers (2 for Improvement and 4 for Averted Loss) are common settings in existing policy and practice, and indeed the averted loss multipliers are based on what is actually used in the two of our case study regions with current operating policies (P6 L177–179). A pristine biodiversity state was used as a simplifying assumption within 'natural vegetation' categories, given that data on vegetation condition was not available at the extent and resolution required for our analysis (P15 L595–596). A 50% restoration success rate was based on evidence from restoration ecology literature (P6 L180–181; P17 L626–634). Further, we quantify and discuss the multipliers needed to achieve NNL in each case study region (Figure 3; P6 L191–192; P9 L336–339).

33. What is really difficult to understand for me is why these parameters have not been calibrated from data collected in the field. Indeed, we could expect to have an empirical analysis based on field work and not only a modelling work for this type of article.

RESPONSE: In addition to justifying model parameters (P6 L177–179; P15 L592–593; P6 L180–181), discussing their implications for results (P17 L622–628) and quantifying the multipliers required to achieve NNL of biodiversity (Figure 3; P6 L191–192; P9 L336–339), we have added R2's point that parameters could be informed by field work. This is particularly true for the restoration success rate of offsetting projects, which should be a key research priority (P16 L631–632). To our knowledge, these data were not available for any of our case studies, and undertaking field work across multiple large scale case studies was beyond our study scope; hence our reliance on published reviews.

34. At the end I even did not really understand why it was necessary to have « real » case studies in this paper, except to provide basic information on specific development projects. In addition no sensitivity analysis has been carried out to assess how the results depend on these arbitrary assumptions. Unfortunately it leads to have both scientific limitations of a broad but rough comparative analysis using big database (lack of qualitative analysis) and of an in-depth fieldwork based on few study sites (lack of quantitative comparisons)... without the strengths of these methods.

RESPONSE: We used realistic case studies to examine the influence of policy design settings and local conditions on policy performance. Ex-ante examination of the large-scale outcomes of different policy settings does unfortunately rely upon a simulation approach, albeit with parameters drawn from real data and real context as much as was possible. We could have conducted a similar analysis using a synthetic landscape and systematically altering local conditions to measure potential policy performance via a sensitivity analysis. However, prior to conducting our analysis, it was not clear which local conditions would have been interesting to examine with such an analysis. Instead, we have added a discussion of how differences in revegetation success may affect results (P16 L626–636) and emphasised the multipliers needed to achieve NNL of biodiversity (Figure 3; P9 L336–339).

35. I would have suggested to start from a big database regarding this topic (for example the IUCN database regarding the compensation policies around the world) and to create some generic categories of countries having common policy designs (using the 18 compensation policy designs options or others). The IUCN database is mentioned (once) but not used at all, which seems surprising. With such an approach, it would have been possible to draw a broad picture of the different level of compliance regarding compensation around the world and to go further in the detail next.

RESPONSE: The 18 policy design options do reflect those commonly used around the world (and therefore included in the GIBOP data base). We have added this point to our manuscript, citing a recent analysis of the GIBOP data (Bull & Strange 2018; P3 L99). However, using the GIBOP database to categorise policy types and then using this to examine compliance is quite a different objective to that achieved by our study. To address R2's comment, we clarify that to achieve our objective, it was necessary to examine a common set of policy options under varied local conditions (P3 L99), specify why our case studies were chosen (P12 L428), and provide a summary of them in a new Table (Table 1).

36. Figure 2 (which summaries the main results of the paper) says three simple things for someone interested in improving ecological compensation methods. First, the no net loss is not reached even in the best policy configuration. Second, it is necessary to have an « improvement approach » since the ecological benefits are always higher than for the « averted loss ». Third, there is no difference between trading types and prioritisation types if you adopt an « improvement approach ». The first result is not surprising since the “arbitrary choices” adopted for calibrating the models are all based on the idea that the compensation does not work. For example, why assuming that the success rate of « improvement » is 50%, without mentioning any source for supporting this assumption ? It is necessary to have, at least, a literature review mentioning this rate. Another option could have been to mention previous results (regarding this rate) in the study sites. Another "arbitrary choice" is that the impacted native vegetation was in its pristine state. With the adoption of such standards, it is not surprising that one of the conclusion of the paper was that the compensation measures don't allow to reach a no net loss goal. Again the main problem is not that these assumption are wrong or right, but that no fieldworks have been carried out to check them (even with a very rapid assessment method).

RESPONSE: Thank you for these suggestions for clarification. We have justified our use of these parameter values (compensation multipliers on P6 L177–179), restoration success rates on P17 L626–636 and vegetation condition on P15 L595–596), to clarify that they are not arbitrary and instead are based in real policy and evidence reviewed in the literature. We have also expanded our discussion around how these values likely affect policy performance. Further, we are clear that our methods quantify “potential” policy performance (P2 L45; P2 L48; P3 L92) of ‘hypothetical [though realistic] compensation policies’ (P3 footnotes) and have reiterated that our findings should not be

used to develop specific policies in our case study regions, but instead used to guide high-level policy development and inform more specific analyses in each of our cases (P16 L650–652).

37. The second result highlighted by Figure 2 (improvement better than averting) is so obvious that it seems not necessary to have a modelling process like this one to say that.

RESPONSE: Interestingly, Improvement will not always outperform Averted Loss, and we show the circumstances under which this is the case. We have now clarified this. While Improvement did achieve better outcomes than Averted Loss in three out of our four case studies (East Kalimantan was the exception, where both approaches gave similar results), our findings are highly dependent on the differences in multipliers between approaches (as stated on P6 L170–172). For this reason, we limited our comparisons to within each approach and discussed the multipliers needed to achieve NNL of biodiversity (P6 L188–191). We have now also highlighted situation where Averted Loss approaches would outperform Improvement – i.e. when rates of restoration success were low, and counterfactual biodiversity gains were high and losses low (P6 L187–189).

38. The third result lead to consider that all this modelling process does not allow to disentangle the parameters to target for improving the success of offsetting policies (only in Cabo Delgado it is possible to highlight that « in kind » approach is worse than others). In conclusion these results cannot be considered as very useful for managers.

RESPONSE: We have sought to clarify further the different policy settings that perform better under different circumstances. The third result highlighted by R2 in comment #36 “there is no difference between trading types and prioritisation types, if you adopt an Improvement approach” is also not always true. As highlighted by our findings in Cabo Delgado, and now explained in our manuscript, in-kind approaches are limited in contexts where insufficient opportunities exist to undertake compensation, or when the vegetation types impacted by development are not threatened by unregulated sectors (P6 L206–208).

Further, to address R2’s concern that our results may not be useful, we have added text to our Conclusion to explicitly state how managers and policy makers can utilise our findings: *“Averted Loss approaches may outperform Improvement approaches when counterfactual biodiversity losses are high and counterfactual gains low, so long as sufficient land exists. In-Kind trades achieve greater gains for the specific vegetation types impacted by regulated development, but will perform similarly well to Out-of-Kind trades when threats are distributed even among types. In general, prioritising compensation near existing protected areas will achieve greater gains in regions where these sites are most threatened.”* (P10–11 L395–401).

39. It was also not clear for me why it is necessary to assess the impact of compensation on two ecosystem services, neither why it was these two ecosystem services which have been chosen, and what is the added value of this information for this paper. I understood that it helps to question the trade-offs but since the starting point of the paper is the “regulated development” (with no legal requirements for the loss of ecosystem services) the ecosystem service section appears a little bit as off-topic. In summary I am sorry to write that the results are not very convincing, neither very exciting.

RESPONSE: We examined the impacts of compensation on ecosystem services because policies increasingly require conservation of biodiversity without negatively affecting people (P8 L295–296), yet no study has examined the extent of potential trade-offs under various policy designs and local

conditions (see edits on P8 L299–301). We also now explain that these two ecosystem services – carbon storage and sediment retention – were chosen because: compensation activities affect these services, they are of value to the communities in all four of our case studies, as highlighted in previous research, and we had access to sufficient data and models to quantify them (P8 L305–308).

Comments about the conclusion section:

40. One of the core message of the paper is that the NNL goal is difficult to reach because there are not enough land available. Unfortunately it is not obvious, for the reader, to understand how the availability of the land is defined in the paper. I guess that it was the surface of vegetation in bad condition and the surface of the cleared land which allow to defined it. If it is the definition retained it can be highlight that other types of landuse could be eligible for compensation, even some built areas. What is sure it that a core issue is not mentioned in the paper regarding the availability of the lands. Indeed, the availability of land is at a given price. Higher is the price you are willing to pay for buying a land which will be used for compensation higher are the available lands for compensation. Indeed, the problem of land availability is, at a certain extent, only a problem of price of the lands. If the price of the compensation is very high (as in certain American states) it is possible to buy some lands where there are buildings for examples, and restore these lands with a great ecological benefit.

RESPONSE: Thank you for pointing out the potential lack of clarity. Given that this definition is crucial for our results, we have now clarified it in our Abstract (P2 L46), Introduction (Table 1 caption) and Methods (P13 L491–502). We had previously defined the extent of land available for compensation in the Discussion (P6 L168–170): “*Averted Loss (i.e. protecting existing vegetation and averting counterfactual losses) and Improvement (restoring and protecting land currently void of vegetation) approaches to generating biodiversity gains*”; and on P8 L236–238 we define the land available for compensation as “*unprotected vegetation to protect for Averted Loss, and cleared land to restore for Improvement*”.

We did not permit revegetation on land that was “built up” with urban or industrialised areas. We have clarified this point and explain that built up areas are unlikely to be restored for compensation, given their relatively high costs to acquire and, even if they were acquired, restoration success of such degraded sites is highly uncertain and may also lead to a displacement of built up areas elsewhere, which would require additional compensation (P13 L493–495).

41. Another limit of the conclusion is that there are no discussion regarding other works carried out on this topic in order to compare the results and discuss/interpret potential differences.

RESPONSE: Our Discussion now refers to previous work on compensation policies, specifically those that evaluate their success. Indeed, our edits to this version of the manuscript have led us to include reference to 16 new papers, many of which undertake such research. However, our study is the first to systematically evaluate the potential performance of policy designs across a diverse set of large-scale case studies (now clarified on P3 L93–96) and we reserved the Conclusion for drawing conclusions based on our results and their relevance to global trends in compensation policies.

Comments about the materials and methods section:

42. The step 1 is OK for me. The step 2 is far to be so convincing for me. First the 18 policy design options are assumed to be « currently used around the world ». OK but it is necessary to give some references to feed this assumption. Next, the reader would like to know a little bit more about the « required amount of compensation » and the « required configuration » but also the

« model developed in Dinamica EGO ». In fact all the sentence gives the feeling to the reader that the analysis is based on a « black box » impossible to understand if you are not involved in the work.

RESPONSE: We have included references to support our statement that the 18 policy design options are frequently used to compensate impacts of development on biodiversity (P13 L484) and have added more text to detail how compensation is allocated to the landscape (P13 L491–502).

43. I don't understand the step 3. The first sentence of the paragraph is « we quantified impacts of compensation (see Step 4) relative to... ». It is a little bit confusing for the reader to try to understand the step 3 starting with the notification that it is necessary to read the step 4 before (assuming that the step 4 is after the step 3...). The following sentences are not more clear for me. I am sorry to write that after three readings I was still completely lost. I was unable to understand, at the end, what was concretely the « counterfactual biodiversity losses and gains ». So the step 4 was not easier to understand. Maybe it is because I am not enough familiar with this type of models. But I suspect it is very challenging to understand this paragraph even if you are familiar with this type of model.

RESPONSE: We edited to the first sentence of “Step 3”, to remove its dependence on “Step 4” and have explicitly defined counterfactual biodiversity losses and gains (P13 L506–509). We have also clarified that the counterfactual scenarios simulated by our land use change model include vegetation loss (to indicate biodiversity losses) and revegetation (to indicate biodiversity gains) (P13 L506–509).

Comments about the assumptions and limitation section

44. What is very frustrating in the limitations mentioned by the authors is that some of them could have been fixed with a two weeks field works. Clearly, it can be expected from this kind of transversal analysis, based on only 4 local case studies, that a real fieldwork was carried out in each of these sites.

RESPONSE: We do disagree with R2 that examining four large case study regions in the field would require only two weeks of work – and of course, most of the scenarios we explore are hypothetical, ex-ante evaluations, and cannot be empirically investigated yet. We suggest that for the objectives of this paper, field work (certainly of the sort which could be quickly carried out) is not an ideal approach. This is because there is value in using consistent large scale data sets to allow inferences about how different policy settings behave, under particular (realistic) assumptions – this allows generalities and principles to emerge. Of course on the ground understanding of each case study is vital to ensure the assumptions are realistic and plausible, and this understanding is achieved through our diverse authorship team.

Reviewer #3 (Remarks to the Author):

45. This study takes up the increasingly prominent notion of "no net loss" conservation policies and uses a modeling approach to test whether and which policy options and different local conditions come close to achieving "no net loss" of biodiversity outcomes. Interestingly, not a singly policy option in not one of the four study areas in different continents seems able to fulfill the "no net loss" promise.

The research question addressed is very timely, the methodology seems robust, and the insights are interesting and relevant to a broad range of target audiences. Overall the paper is well-written and structured. But as it contains a lot of content and many potentially valuable insights, I'd urge the authors to make the paper as clear and understandable as possible.

RESPONSE: We thank R3 for their comments and constructive feedback on our manuscript. Below we respond to each, illustrating how our manuscript is now much improved in clarity. Please note that page and line numbers quoted in our responses refer to the tracked changes version of our revised manuscript.

46. In particular - while I find the results and discussion on policy design clear - I am struggling a bit with the part on local conditions (availability of land and counterfactual biodiversity losses and gains). Here, I feel the reader needs more explanation. How was the amount of available land for compensation defined? Within which geographic boundaries? A clearer definition with examples of counterfactual losses and gains may also be helpful.

RESPONSE: We have revised our manuscript for clarity and added a new table to summarise differences in local conditions among case studies (Table 1). We have also added definitions of the 'land available for compensation' earlier in the manuscript (P2 L49–50; Table 1 caption) and provided examples of what counterfactual losses and gains represent – i.e. the extent of vegetation types was our proxy for biodiversity (P3 L110) and simulated unregulated changes in vegetation extent represent counterfactual losses and gains (P3 L110; P13 L506–511).

47. As for the policy design - what I read from Figure 2 is that "Improvement" options generally perform much better than "Averted Loss" options. And that within the "Improvement" category, most trade and prioritisation options perform similarly in the four study areas. Shouldn't a stronger policy message be taken out of that?

RESPONSE: Our set multipliers turned out to be key in explaining differences between Averted Loss and Improvement approaches (P6 L170–172). For this reason we have limited our comparisons of performance to within approaches (as in, within Averted Loss, and separately, within Improvement) and have also calculated and discussed the multipliers required to achieve>NNL of biodiversity given the scenarios we explored for each (P6 L191; P9 L339). We now also highlight the situation where Averted Loss would outperform Improvement – i.e. when rates of restoration success are low, and counterfactual biodiversity gains were high and losses low (P6 L187–189).

Similarly, we have clarified the situation where some difference could be expected among types of trades for Improvement approaches. As highlighted by our findings in Cabo Delgado, and now explained in our manuscript, In-kind approaches are limited in contexts where insufficient opportunities exist to undertake compensation, or when the vegetation types impacted by development are not threatened by unregulated sectors (P6 L206–208).

Finally, we have summarised the policy messages of our findings in the Conclusion (P10–11 L395–399).

48. The additional consideration of two ecosystem services adds value to the manuscript. While it is intuitive to study additional effects on carbon storage, the consideration of sediment retention appears a bit random. Can you find some stronger arguments of why outcomes of compensation in terms of sediment retention should be considered? Is that really an ecosystem service of outstanding importance in the four study areas / for the types of development studied?

RESPONSE: Thank you for this comment and suggestion. We have clarified why sediment retention, along with carbon storage, were important services to analyse in this study (L8 P305–308). Specifically, sediment retention is provided by the vegetation lost and gained from development and compensation and is of value to the local communities across our case studies (see refs 29–33).

49. The study yields many results that are relevant for national and global conservation policies. I'd like to encourage the authors to distil some stronger policy insights, ideally in the conclusions section. For example, the authors mention twice that further development may have to be prevented once compensation options are exhausted. This could become a strong policy message. Also, shouldn't your results give reason to question the usefulness of the term "no net loss" of biodiversity - given that not a singly of your options achieved no net loss? Is NNL no more than a buzzword then?

RESPONSE: As suggested, we expanded our Conclusion to highlight policy insights obtained from our analysis (P10–11 L395–399) and reiterated more strongly the importance of avoiding impacts (P10 L412–413). We have also specified in our Discussion that to achieve NNL of biodiversity, development impacts must cease once compensation options are exhausted (P7 L253–255).

Minor remarks:

50. L. 128-130: I have difficulties understanding the justification in the note. Consider rewording / clarifying.

RESPONSE: We have reworded this sentence for clarity. *"Note: we only prioritised compensation Within PAs for Out-of-Kind trades, given that protected areas had limited opportunities to undertake compensation (they did not contain much cleared land for Improvement approaches, or experience high rates of counterfactual losses for Averted Loss approaches) and investigating even more restrictive 'in-kind' trades would have further reduced their impact"* (P5 L158–161).

51. L. 200-201: Something seems missing in this sentence.

RESPONSE: We apologise and have corrected this typo. The sentence now reads: *"there was insufficient unprotected vegetation to protect for Averted Loss, or cleared land to restore for Improvement, to allow compensation for all regulated development simulated over our investigated timeframe"* (P7 L236–238).

Reviewers' comments:

Reviewer #1 (Remarks to the Author):

The authors were very responsive in their revisions and fully addressed all my comments. The manuscript is much improved and is the first to systematically examine how local conditions interact with policy design to affect the potential performance of biodiversity compensation schemes across world sites. The methods are well documented, and the modeling approach seems sound. I believe that the findings – particularly the apparent major limitations of compensation policies at achieving their goals when applied to entire regional industries, but also the practical suggestions on how to improve outcomes by incorporating consideration of local conditions – will be of great interest to conservation scientists and the broader conservation practitioner community.

Reviewer #2 (Remarks to the Author):

Thank you for the response to my comments.
Here are my new comments.

Response 31 : Sorry but I am still not convinced by the figure 5. As a social scientist it is difficult to consider this figure as a theoretical framework usable for analysing different policy design.

Response 32 : There is a strong problem regarding this response. It is mentioned that « A 50% restoration success rate was based on evidence from restoration ecology literature ». As far as I know this rate is wrong. For example, Benayas and al. (2009) is mentioned as one of these "evidence from restoration ecology literature". But in this publication the rate of success (in comparison with a pristine state) is 86% and not 50%. Maybe I have missed something in this publication, but I don't think so. Only one additional publication (Suding et al., 2011) is cited, with Benayas et al., to justify the 50%, while there are more recent publications on this topic such as, for example, Jones, H.P., Jones, P.C., Barbier, E.B., Blackburn, R.C., Rey Benayas, J.M., Holl, K.D., McCrackin, M., Meli, P., Montoya, D., Mateos, D.M., 2018. Restoration and repair of Earth's damaged ecosystems. *Proceedings of the Royal Society B: Biological Sciences* 285, 20172577. It could also be interesting to mention specific rate of success for specific habitats. For example : Moreno-Mateos, D., Power, M.E., Comin, F.A., Yockteng, R., 2012. Structural and Functional Loss in Restored Wetland Ecosystems. *PLoS. Biol.* 10, e1001247.). In this publication, the rate of success of restoration project for wetlands is around 75% both for functional and structural biodiversity (depending mainly on the size of the restoration project). To be short, I am still not convinced by the rate of 50%. And I guess this rate influences a lot the results presented in the paper.

Responses 33 to 35 : I understand that data are missing and that fieldwork was not feasible. But in this case I still don't understand why there is a focus on few case-studies and not on a bigger sample of countries/areas where the variables required for the analysis would be available.

Responses 36 to 38 are OK for me.

Response 39 is OK for me, even if the carbon storage and the sediment retention are not the ecosystem services which are the most important for local population impacted by offset projects.

Response 40 clarifies the hypothesis on what is named « land available », but the economic dimension of this variable is missing and it is a clear weakness of the model. Indeed, a basic economic law is that if the demand for lands (for offsetting projects) increases, and the supply of land is steady, then the price of the land as well as the cost of compensation increase too. It quickly leads to have incentives for avoiding and mitigating impacts, and to decrease the demand for lands. Considering that the model address the problem of land availability for offsetting policies, missing this basic economic law is clearly an important limit of the model. Also, considering the built areas as ineligible for compensation is questionable since strong ecological lift can be obtained from these areas, even if it is impossible to reach a pristine state. In addition,

even if the costs of these lands are high, it is again a question of ratio between the level of demand and the level of supply, and this high cost can be included in the price of the compensation.

Responses to comment 42 to 44 are OK for me.

Reviewer #3 (Remarks to the Author):

I think the author have done a very good revision, and the manuscript has gained a lot of clarity. My original issues have all been addressed sufficiently, and I now recommend to accept the paper.

Reviewers' comments:

Reviewer #1 (Remarks to the Author):

1. The authors were very responsive in their revisions and fully addressed all my comments. The manuscript is much improved and is the first to systematically examine how local conditions interact with policy design to affect the potential performance of biodiversity compensation schemes across world sites. The methods are well documented, and the modeling approach seems sound. I believe that the findings – particularly the apparent major limitations of compensation policies at achieving their goals when applied to entire regional industries, but also the practical suggestions on how to improve outcomes by incorporating consideration of local conditions – will be of great interest to conservation scientists and the broader conservation practitioner community.

RESPONSE: Thank you for your feedback.

Reviewer #2 (Remarks to the Author):

Thank you for the response to my comments. Here are my new comments.

RESPONSE: Thank you for these comments. Below we respond to each, indicating where our manuscript has been edited accordingly. Please note that page and line numbers refer to our edited manuscript when viewing in-line tracked-changes (see pdf version of revised manuscript).

2. Response 31: Sorry but I am still not convinced by the figure 5. As a social scientist it is difficult to consider this figure as a theoretical framework usable for analysing different policy design.

RESPONSE: Our study is focused on examining how the technical settings of policies (i.e. the compensation approach used, the types of trades, and how offsets are prioritised across the landscape) and the regulated development footprint affect biodiversity. We are not employing social science methods, nor are we examining socio-ecological dynamics. As discussed in the manuscript (see P2 L86–89), many other factors will affect policy design, implementation and success in reality, including socio-ecological dynamics; however, these all fall outside our study scope.

3. Response 32: There is a strong problem regarding this response. It is mentioned that « A 50% restoration success rate was based on evidence from restoration ecology literature ». As far as I know this rate is wrong. For example, Benayas and al. (2009) is mentioned as one of these “evidence from restoration ecology literature”. But in this publication the rate of success (in comparison with a pristine state) is 86% and not 50%. Maybe I have missed something in this publication, but I don't think so. Only one additional publication (Suding et al., 2011) is cited, with Benayas et al., to justify the 50%, while there are more recent publications on this topic such as, for example, Jones, H.P., Jones, P.C., Barbier, E.B., Blackburn, R.C., Rey Benayas, J.M., Holl, K.D., McCrackin, M., Meli, P., Montoya, D., Mateos, D.M., 2018. Restoration and repair of Earth's damaged ecosystems. Proceedings of the Royal Society B: Biological Sciences 285, 20172577. It could also be interesting to mention specific rate of success for specific habitats. For

example : Moreno-Mateos, D., Power, M.E., Comin, F.A., Yockteng, R., 2012. Structural and Functional Loss in Restored Wetland Ecosystems. PLoS. Biol. 10, e1001247.). In this publication, the rate of success of restoration project for wetlands is around 75% both for functional and structural biodiversity (depending mainly on the size of the restoration project). To be short, I am still not convinced by the rate of 50%. And I guess this rate influences a lot the results presented in the paper.

RESPONSE: We agree with Reviewer 2 – restoration success rates likely vary considerably among places, ecosystems, modes of restoration and types of initial disturbance. We have clarified this point on P16 L617–618, citing the suggested references of Jones et al. 2018. In some cases, success rates could greatly exceed 50%, if initial conditions are favourable and manage regimes strict and effective. In other cases, success rates may fall far below 50%, as shown by Jones et al. (2018) and others (e.g. van Katwijk et al. 2016), which also found that active restoration rarely perform significantly better than passive recovery. Again, in other cases, passive restoration performs better (Crouzeilles et al. 2017, Shimamoto et al. 2018).

Based on the studies cited above, we argue that our 50% success rate is a reasonable, although a probably optimistic, one to apply for illustrative purposes across the case studies. However, to address Reviewer 2's comments and to test the sensitivity of our results to our chosen success rate, we have further explored the implications of a lower (25%) and higher (75%) success rate (see new Fig S4). We discuss the implications of these alternative rates in the manuscript. Specifically, as restoration success increases (while multipliers remain constant), so too does the impact of compensation on biodiversity and, in some cases, achieves>NNL (P16 L612–615). However, it is worth noting that if restoration success does increase, policy multipliers will also be reduced to reflect this (P16 L619–620).

4. Responses 33 to 35: I understand that data are missing and that fieldwork was not feasible. But in this case I still don't understand why there is a focus on few case-studies and not on a bigger sample of countries/areas where the variables required for the analysis would be available.

RESPONSE: Our modelling required significant data inputs, many of which are not readily available at broad spatial scales. Analysing and interpreting results also required in-country expertise. For these reasons, we have chosen 4 detailed case studies (informed by best available data, local knowledge and chosen to represent a range of contexts). While adding more case studies to our study may have provided additional insights, it is unlikely to have changed the findings substantially.

5. Response 40 clarifies the hypothesis on what is named « land available », but the economic dimension of this variable is missing and it is a clear weakness of the model. Indeed, a basic economic law is that if the demand for lands (for offsetting projects) increases, and the supply of land is steady, then the price of the land as well as the cost of compensation increase too. It quickly leads to have incentives for avoiding and mitigating impacts, and to decrease the demand for lands. Considering that the model address the problem of land availability for offsetting policies, missing this basic economic law is clearly an important limit of the model. Also, considering the built areas as ineligible for compensation is questionable since strong ecological lift can be obtained from these areas, even if it is impossible to reach a pristine state. In addition, even if the costs of these lands are high, it is again a question of ratio between the level of demand and the level of supply, and this high cost can be included in the price of the compensation.

RESPONSE: We agree with Reviewer 2 that, in theory, increasing demand for land for offsetting while supply remains constant, will increase land prices and thus create incentives for impact avoidance once land prices exceed the returns from development. In fact, many working on offsetting from an environmental perspectives believe one of the values of NNL laws is that the challenges of offsetting pushes developers towards more avoidance. In reality, however, rarely is this dynamic observed in an offsetting context. Instead, when land is scarce, there are many examples of compensation requirements being eased (e.g. less compensation is undertaken; other forms of compensation, such as investment in ecological research, are permitted instead). To address this comment, we have clarified that our offset allocation model does not consider dynamic land prices (P13 L473–474) and explained that this likely aligns with reality (P13 L475–479). We note that, even if land costs are high enough to encourage purchase of built areas for conversion to a natural state (and this seems unlikely, given the above), it is not technically possible to do so within a meaningful timeframe (P13 L480–482).

Reviewer #3 (Remarks to the Author):

6. I think the author have done a very good revision, and the manuscript has gained a lot of clarity. My original issues have all been addressed sufficiently, and I now recommend to accept the paper.

RESPONSE: Thank you for your feedback.